# Listening Through the Noise: Cauchy-Driven Diffusion Bridges for Robust Gastrointestinal Auscultation and Clinical Benchmarking

**Dian Ding**[1]  **Liren Dong**[2]  **Yu Lu**[1]  **Juntao Zhou**[1]  **Ran Wang**[1]  **Peng Li**[2]  **Zhenyi Jia**[3]  **Guangtao Xue**[1]

## Abstract

Gastrointestinal (GI) motility assessment via bowel sounds (BS) offers a non-invasive alternative to resource-intensive clinical standards. However, the diagnostic utility of BS is often compromised by its spectral overlap with non-stationary speech interference. While generative models have advanced signal restoration, traditional Gaussian-based diffusion frameworks struggle with the impulsive, heavy-tailed nature of real-world clinical noise. In this paper, we propose a novel Cauchy-driven Diffusion Bridge framework to isolate high-fidelity bowel sounds from complex interference. Our contributions are three-fold: (1) We introduce CLINBS, a large-scale clinical dataset (over 25 hours) containing rare pathological transients verified by experts; (2) We mathematically formulate a Cauchy bridge driver, deriving closed-form expressions for the score and density to better model heavy-tailed perturbations; and (3) We implement an efficient sampling procedure via Gaussian scale-mixture reparameterization. Extensive experiments show our framework achieves state-of-the-art performance, outperforming baselines by 13.4%–49.8% across core metrics and elevating abnormal BS recognition accuracy to 88.01%. These results demonstrate the system's potential for robust clinical GI monitoring and diagnosis.

## 1. Introduction

Gastrointestinal (GI) motility assessment (Yadlapati et al., 2021) is vital for managing digestive disorders, yet current

[1]School of Computer Science, Shanghai Jiao Tong University, Shanghai, China [2]School of Artificial Intelligence and Computer Science, Shaanxi Normal University, Xi'an, China [3]Department of General Surgery, Shanghai Sixth People's Hospital, Shanghai Jiao Tong University School of Medicine, Shanghai, China. Correspondence to: Yu Lu <yulu01@sjtu.edu.cn>.

*Proceedings of the $43^{rd}$ International Conference on Machine Learning*, Seoul, South Korea. PMLR 306, 2026. Copyright 2026 by the author(s).

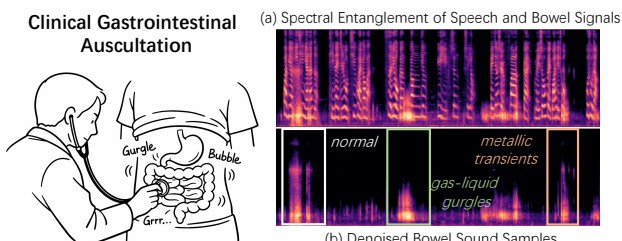

*Figure 1.* Gastrointestinal auscultation is vital for non-invasive motility assessment. Identifying **gas-liquid gurgles** and **metallic transients** is essential for diagnosing mechanical bowel obstruction and guiding acute abdomen interventions. However, spectral overlap from human speech masks key acoustic features of hyperperistalsis, significantly hindering the robust detection of pathological bowel sounds in clinical settings.

gold standards—ranging from imaging to invasive manometry—are episodic, resource-intensive, and unsuitable for longitudinal monitoring. Bowel sounds (BS) provide a non-invasive acoustic proxy for GI motility (Baronetto et al., 2023), with acoustic deviations clinically linked to pathologies like mechanical ileus and inflammatory bowel disease (Yoo et al., 2023) (Fig. 1). Despite their diagnostic potential, bowel sounds are weak, stochastic signals highly susceptible to environmental contamination (Allwood et al., 2018). In clinical settings, recordings are frequently obscured by speech interference from patient–clinician interactions. Critically, the spectral footprint of bowel sounds (100–1000 Hz) overlaps significantly with human speech formants. This spectral entanglement necessitates a generative approach capable of disentangling these complex distributions to ensure 'listening through the noise' without compromising the clinical fidelity of the biological signal.

The restoration of high-fidelity physiological signals—such as EEG, ECG, and bowel sounds—remains a formidable challenge due to the prevalence of non-stationary artifacts and the inherent sparsity of clinical transients. Conventional noise suppression techniques (Boll, 2003) typically rely on spectral subtraction, which often fails to distinguish target signals from structured interference. While modern discriminative architectures—ranging from early U-Net adaptations (Perslev et al., 2019) to state-of-the-art backbones like TimesNet (Wu et al., 2023) and Mamba (Gu & Dao,

2023)—have significantly advanced feature representation, they are fundamentally tethered to deterministic regression objectives. To overcome these constraints, the field has increasingly pivoted toward generative frameworks, such as Diffusion Models and Schrödinger Bridges (Liu et al., 2023), to recover the underlying clean signal manifolds. However, existing generative formulations predominantly operate under Gaussian noise priors (Song et al., 2021), a simplification that is fundamentally misaligned with the impulsive, heavy-tailed nature of real-world clinical perturbations (Simsekli et al., 2019; Lian et al., 2025).

Beyond the theoretical constraints of noise modeling, the practical deployment of gastrointestinal (GI) auscultation systems faces two critical systemic challenges.

**First, there is a severe scarcity of high-fidelity clinical datasets for bowel sounds.** Current benchmarks fail to encompass the full acoustic diversity of GI motility, particularly the gas-liquid gurgles and metallic transients characteristic of pathological states. This lack of diverse, annotated clinical data severely limits the capacity to evaluate and validate the efficacy of denoising frameworks in authentic clinical environments.

**Second, existing frameworks struggle to disentangle bowel sounds from highly structured, non-stationary interferences, most notably ambient speech.** Unlike stochastic noise, speech signals exhibit harmonic structures and temporal patterns that overlap significantly with the frequency bands of bowel motility bursts. Over-smoothing often blurs the sharp spectral peaks and high-frequency transients that define distinct intestinal conditions.

In this paper, we leverage a Diffusion-Bridge-based generative model and Cauchy-distributed priors to isolate bowel sounds from complex clinical gastrointestinal auscultation environments.

**First, we constructed a large-scale clinical bowel sound dataset (CLINBS).** This dataset consists of 1531 samples, totaling over 25 hours of recordings, and covers a broad spectrum of gastrointestinal diseases, including abdominal infections, intestinal obstruction, and reduced gut function. In addition to standard bowel sounds, it uniquely includes and annotates rare pathological samples that are often missing from existing benchmarks, such as 290 cases of gas-liquid gurgles and 66 cases of metallic transients. To ensure high-quality labeling, the dataset's annotations were rigorously cross-verified by multiple gastroenterologists.

**Second, we propose a Cauchy-driven diffusion bridge framework.** Unlike conventional diffusion models that rely on Gaussian perturbations and a fixed standard Gaussian terminal prior, our approach in Fig. 3 directly targets paired clean-to-noisy restoration by replacing the Gaussian bridge driver with a Cauchy driver. This design is motivated by

our empirical finding that speech interference exhibits pronounced heavy tails, making the Cauchy driver a better match for impulsive and non-stationary corruptions. **Theoretical analysis:** Under standard endpoint and schedule conditions inherited from diffusion bridge models, we show that the proposed construction induces a well-defined heavy-tailed conditional bridge kernel at any intermediate time. We further derive closed-form expressions for the corresponding log-density and score under a factorized Cauchy assumption, and show that the bridge recovers deterministic limits at the endpoints, providing a principled foundation for training and sampling.

**Finally, we develop dedicated training and sampling procedures for the Cauchy-Driven Diffusion Bridge.** For training, we adopt the standard endpoint-prediction parameterization and construct a plug-in score surrogate, which turns conditional-score learning into a tractable endpoint regression problem. Our objective combines three complementary constraints: a task-aligned spectrogram fidelity loss, a Cauchy-kernel-induced heavy-tailed consistency penalty that improves robustness to impulsive outliers, and a spectral structural constraint that preserves bowel-sound textures and frequency co-occurrence patterns. For sampling, we extend diffusion-bridge implicit sampling to the heavy-tailed setting via a Gaussian scale-mixture reparameterization of the Cauchy driver, enabling efficient closed-form DBIM-style refinement updates without reverse-time SDE integration.

Extensive experiments demonstrate that the proposed framework achieves state-of-the-art (SOTA) performance in bowel sound auscultation. In challenging clinical scenarios characterized by non-stationary speech interference, our system yields substantial improvements ranging from 13.4% to 49.8% over the baselines across core metrics including MAE, SSIM, and LSD. Furthermore, the denoising gains directly bolster the reliability of downstream tasks, elevating abnormal bowel sound recognition accuracy by 3.6% to 4.1% compared to baselines. These results validate the framework's capacity to provide robust support for clinical gastrointestinal diagnosis and therapeutic assessment.

**Conflict of Interest Disclosure.** The authors declare no financial conflicts of interest relevant to this work.

## 2. Related Work

### 2.1. End-to-End Signal Separation

To reduce uncertainties in biosignal environments and achieve efficient reconstruction of original signals, signal reconstruction techniques have gradually evolved from traditional data-driven models (Perslev et al., 2019; Peng et al., 2024; Wu et al., 2023) to end-to-end deep learning approaches. Wave-U-Net (Stoller et al., 2018) performs end-to-end mapping from mixed signals to target signals by di-

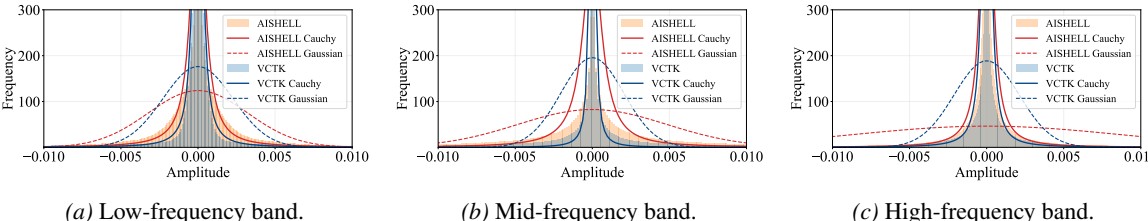

*(a)* Low-frequency band.        *(b)* Mid-frequency band.        *(c)* High-frequency band.

*Figure 2.* Empirical distributions of speech interference noise across different frequency bands showing heavy tails.

rectly encoding and decoding the original waveforms. Conv-TasNet (Défossez, 2021) and Demucs (Défossez et al., 2019) enhance separation quality while maintaining phase consistency through learnable encoder-decoder architectures. Methods such as Demucs-Hybrid and Mel-RoFormer (Wang et al., 2024) combine frequency-domain features with Transformer models and leverage self-attention mechanisms to strengthen long-term dependency modeling. SCNet's (Tong et al., 2024) band-sparsity modeling strategy significantly improves separation accuracy and perceptual quality. However, these methods are often prone to residual noise or artifacts, fail to recover important signal details.

### 2.2. Generative Modeling and Robust Priors

To preserve these subtle biological textures, the field has increasingly pivoted towards generative modeling. Recent works have successfully employed GANs (Pascual et al., 2017; Yoon et al., 2019) and, more significantly, Diffusion Probabilistic Models (DPMs) (Tashiro et al., 2021) to capture the complex manifold of clean physiological signals.

While standard DPMs generate samples from pure Gaussian noise, recent Schrödinger Bridge (Liu et al., 2023) and geometric bridge (Huang et al., 2024) have been introduced to establish a direct transport path between the noisy and clean distributions, enhancing both consistency and efficiency. However, both standard DPMs and existing Bridge formulations typically assume Gaussian noise priors (Song et al., 2021). This assumption is fundamentally misaligned with the heavy-tailed nature of real-world clinical artifacts (e.g., impulsive speech interference) (Simsekli et al., 2019), necessitating a robust formulation that extends beyond Gaussian statistics. While Cauchy Diffusion (Lian et al., 2025) has validated the feasibility of leveraging the heavy-tailed properties of the Cauchy distribution, establishing a Diffusion Bridge based on Cauchy noise still lacks sufficient theoretical foundation and empirical verification.

### 3. Preliminaries

We first investigate the statistical nature of speech noise in bowel sound recordings to motivate a heavy-tailed noise model. Consider the observed mixture $y(t)$ of clean bowel sounds $x(t)$ and additive noise $n(t)$ (primarily speech inter-

ference), modeled as: $y(t) = x(t) + n(t)$. where $n(t)$ is typically assumed Gaussian in conventional methods. This Gaussian assumption is often mismatched for real-world speech noise, which is known to produce bursty, large-amplitude outliers (e.g., sudden loud words or coughs). We suspect that speech noise has a heavy-tailed distribution, meaning it generates extreme values far more frequently than a Gaussian would predict. To examine the statistical nature of speech interference, we conducted a preliminary empirical study using a large speech dataset as the noise source. We segmented hours of speech recordings of both the AISHELL (Fu et al., 2021) and VCTK (Yamagishi et al., 2019) datasets into short frames and applied band-pass filtering to each frame to obtain low-, mid-, and high-frequency components. For each band, we measured the frame-wise loudness and aggregated these measurements over the entire dataset to form empirical distributions (Fig. 2). The resulting distributions reveal a sharp peak around zero but exhibit heavy tails, indicating that speech interference is typically quiet yet occasionally produces large-magnitude bursts. This heavy-tailed behavior motivates modeling the interference with heavy-tailed noise, rather than a Gaussian assumption.

We quantitatively compared a Gaussian fit and a heavy-tailed Cauchy fit for each frequency band's noise distribution. The Cauchy distribution, defined as

$$p(n_i) \approx Cauchy(n_i | \mu_i, \gamma_i) = \frac{1}{\pi \gamma_i [1 + (\frac{n-\mu_i}{\gamma_i})^2]} \quad (1)$$

was found to model the data far more accurately than a Gaussian. Here $\mu_i$ and $\gamma_i$ are the location and scale parameters of the Cauchy in band $f_i$. Notably, the fitted location parameters were $\mu_i \approx 0$ across all bands, confirming that the noise in each band is zero-mean and symmetric. Fig. 2 illustrates a representative frequency band's empirical histogram with the fitted curves: the Cauchy distribution closely tracks the empirical heavy-tailed shape, whereas the Gaussian falls off too quickly in the tails. In particular, the Cauchy fit assigns much higher probability to large noise amplitudes (impulsive events) than the Gaussian fit. This means the Cauchy model better captures rare but significant noise bursts present in speech interference. This motivates our Cauchy-driven diffusion bridge, which matches

the empirical heavy-tail behavior and better accommodates occasional large bursts during restoration.

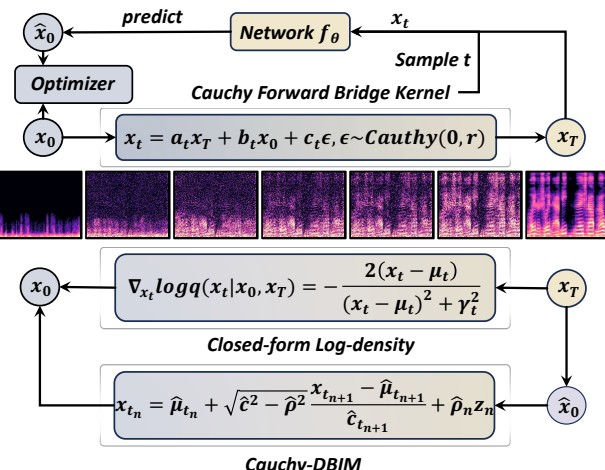

*Figure 3.* Overview of the Cauchy-Driven Diffusion Bridge framework. **Training**: sample $t$, generate $x_t$ with a Cauchy forward bridge, and predict $\hat{x}_0$ to optimize the objective. **Sampling**: starting from $y$, Cauchy-DBIM uses a Gaussian scale-mixture reparameterization for fast closed-form bridge-implicit updates.

# 4. Methodology

## 4.1. Diffusion Bridge Model

Diffusion models (Sohl-Dickstein et al., 2015; Ho et al., 2020; Song et al., 2021) specify a known forward noising process that maps data to a simple reference distribution, learn a reverse-time denoising direction, and generate samples by integrating the reverse SDE (or probability-flow ODE) from $t = T$ back to $t = 0$ (see App. A.1).

A key limitation of standard diffusion is that it typically adopts a non-informative Gaussian prior at $t = T$, hence transporting data toward a fixed standard Gaussian rather than translating between two arbitrary endpoint distributions. In paired restoration problems, however, one is given an informative endpoint $y$ (the observation), and wishes to sample $x_0$ conditioned on $x_T = y$. Denoising Diffusion Bridge Models (DDBM) (Zhou et al., 2024) address this by conditioning the diffusion to hit $x_T = y$ via Doob's $h$-transform:

$$
\begin{aligned}
\mathrm{d}x_t = f(x_t, t)\mathrm{d}t &+ g^2(t)\nabla_{x_t}\log q(x_T = y \mid x_t)\mathrm{d}t \\
&+ g(t)\mathrm{d}w_t, \quad (x_0, y) \sim p_{data}(x, y),\ x_T = y.
\end{aligned}
\tag{2}
$$

With appropriate schedules, the resulting bridge admits a tractable Gaussian conditional kernel,

$$
q(x_t \mid x_0, x_T = y) = \mathcal{N}(a_t y + b_t x_0,\ c_t^2 I),
\tag{3}
$$

equivalently $x_t = a_t x_T + b_t x_0 + c_t\varepsilon,\quad \varepsilon \sim \mathcal{N}(0, I)$, where $(a_t, b_t, c_t)$ are fully determined by the chosen noise schedule (App. A.2).

## 4.2. Cauchy-Driven Diffusion Bridge Model

### 4.2.1. CAUCHY FORWARD BRIDGE KERNEL

Sec. 3 shows that, speech interference is distinctly heavy-tailed: a Cauchy fit matches the empirical tails much better than a Gaussian fit (Fig. 2). Since the fitted interference is often centered (e.g., $\mathrm{Cauchy}(0, \gamma_i)$) and both the centered Cauchy and the standard Gaussian are strictly $\alpha$-stable distributions (West et al., 1997) (App. A.3.1), we are naturally motivated to modify DDBM by replacing the Gaussian bridge noise with heavy-tailed Cauchy noise to better model impulsive real-world corruptions. Specifically, we redefine the forward process as

$$
x_t = a_t x_T + b_t x_0 + c_t\varepsilon, \qquad \varepsilon \sim \mathrm{Cauchy}(0, r).
\tag{4}
$$

Here $x_0$ denotes the clean bowel sound, $x_T \equiv y$ is the observed corrupted signal, and $r$ is the Cauchy scale. Before developing the reverse-time dynamics, we first prove that the induced bridge kernel $q(x_t \mid x_o, x_T)$ is well-defined; we then provide its closed-form log-density and score in the next subsection.

**Proposition 4.1** (Existence of the Cauchy bridge kernel, proof in App. A.3.2). *Assume the schedules satisfy the same endpoint conditions as DDBM for Eq. 4. Then the conditional distribution $q(x_t \mid x_0, x_T)$ exists and is a (multivariate) Cauchy with location $\mu_t = a_t x_T + b_t x_0$ and scale $c_t r$, i.e.,*

$$
q(x_t \mid x_0, x_T) = \mathrm{Cauchy}\big(x_t \mid \mu_t,\ c_t r\big).
$$

*Moreover, $x_0$ and $x_T$ are satisfied almost surely by the endpoint conditions.*

The proposition implies that the Cauchy-driven construction yields a valid heavy-tailed bridge forward kernel $q(x_t \mid x_0, y)$ for any intermediate time $t \in (0, T)$ (with degenerate endpoints).

### 4.2.2. CLOSED-FORM LOG-DENSITY AND SCORE

For both training and sampling, we require the log-density and the score (gradient w.r.t. $x_t$) of the Cauchy bridge kernel. The following result provides closed-form expressions.

**Proposition 4.2** (Closed-form log-density and score of the Cauchy bridge kernel, proof in App. A.3.3). *Under the setting of Proposition 4.1, for any $t \in (0, T)$ the conditional density $q(x_t \mid x_0, y)$ is given by Eq. (45) in App. A.3.2 with $\mu_t = a_t y + b_t x_0$ and $\gamma_{t,i} = c_t r_i$. Moreover, its log-density and score (gradient w.r.t. $x_t$) admit closed forms:*

$$
\begin{aligned}
\log q(x_t \mid x_0, y) = &-\sum_{i=1}^{d}\log\big(\pi\gamma_{t,i}\big) \\
&-\sum_{i=1}^{d}\log\Big((x_{t,i} - \mu_{t,i})^2 + \gamma_{t,i}^2\Big),
\end{aligned}
\tag{5}
$$

$$\nabla_{x_t} \log q(x_t \mid x_0, y) = \left( -\frac{2(x_{t,i} - \mu_{t,i})}{(x_{t,i} - \mu_{t,i})^2 + \gamma_{t,i}^2} \right)_{i=1}^{d} \quad (6)$$

Compared to the Gaussian bridge, this tractable heavy-tailed bridge kernel preserves the same endpoint interpolation structure while providing robustness to impulsive corruptions, which is crucial for bowel sound restoration under speech interference.

### 4.2.3. TRAINING OBJECTIVE

Following DDBM, we aim to learn the *marginal* bridge score $s^\star(x_t, t, y) \triangleq \nabla_{x_t} \log q_t(x_t \mid y)$, where $q_t(x_t \mid y) = \int q(x_t \mid x_0, y) \, p(x_0 \mid y) \, dx_0$ is the marginal of the conditional forward bridge kernel $q(x_t \mid x_0, y)$. A natural objective is denoising bridge score matching (BSM):

$$\mathcal{L}_{\text{BSM}}(\theta) = \mathbb{E}_{(x_0, y) \sim p_{\text{data}}} \mathbb{E}_t \mathbb{E}_{x_t \sim q(\cdot \mid x_0, y)}$$
$$\left[ w(t) \left\| s_\theta(x_t, t, y) - \nabla_{x_t} \log q(x_t \mid x_0, y) \right\|_2^2 \right], \quad (7)$$

where $s_\theta$ is the learnable score model and $\nabla_{x_t} \log q(x_t \mid x_0, y)$ is the *conditional* bridge score. In general, $s^\star$ is intractable due to the marginalization over $x_0$. In our Cauchy bridge, however, the conditional score $\nabla_{x_t} \log q(x_t \mid x_0, y)$ admits a closed form (Proposition 4.2), enabling tractable supervision at the conditional level. Instead of directly parameterizing $s_\theta$, we adopt the standard $x_0$-parameterization and predict the clean endpoint

$$\hat{x}_0 = f_\theta(x_t, t, y), \qquad \hat{\mu}_t \triangleq a_t y + b_t \hat{x}_0, \quad (8)$$

which induces a plug-in conditional score by substituting $\mu_t$ with $\hat{\mu}_t$ in Proposition 4.2:

$$\nabla_{x_t} \log q(x_t \mid \hat{x}_0, y) = \left( -\frac{2(x_{t,i} - \hat{\mu}_{t,i})}{(x_{t,i} - \hat{\mu}_{t,i})^2 + \gamma_{t,i}^2} \right)_{i=1}^{d}, \quad (9)$$

where $\gamma_{t,i} = c_t r_i$. This construction connects bridge score learning with endpoint prediction and yields a computable surrogate score model.

**Proposition 4.3** (Endpoint regression as a plug-in bridge parameterization, proof in App. A.3.4). *Let $q(x_t \mid x_0, y)$ be a differentiable bridge kernel and define the marginal $q_t(x_t \mid y) = \int q(x_t \mid x_0, y) \, p(x_0 \mid y) \, dx_0$. Then the marginal bridge score satisfies the Fisher identity*

$$\nabla_{x_t} \log q_t(x_t \mid y) = \mathbb{E}[\nabla_{x_t} \log q(x_t \mid x_0, y) \mid x_t, t, y]. \quad (10)$$

*In addition, the $\ell_2$-risk minimizer for predicting $x_0$ from $(x_t, t, y)$ is $f^\star(x_t, t, y) = \mathbb{E}[x_0 \mid x_t, t, y]$. Consequently, if $\nabla_{x_t} \log q(x_t \mid x_0, y)$ admits a closed form and depends on*

$x_0$ *only through a bridge parameterization (e.g., a location parameter $\mu_t = \mu_t(x_0, y, t)$), one may construct a* plug-in *score model by replacing $x_0$ (equivalently, $\mu_t$) with $\hat{x}_0 = f_\theta(x_t, t, y)$ (equivalently, $\hat{\mu}_t = \mu_t(\hat{x}_0, y, t)$). This yields a computable proxy $s_\theta^{\text{ind}}(x_t, t, y) = \nabla_{x_t} \log q(x_t \mid \hat{x}_0, y)$ for approximating the intractable marginal score in* (10). *The approximation is justified when $p(x_0 \mid x_t, t, y)$ is concentrated, under which a point estimate $\hat{x}_0$ captures the dominant posterior mass.*

**Endpoint fidelity and Cauchy consistency.** Motivated by the plug-in construction, we retain endpoint supervision and further enforce heavy-tailed consistency of the bridge residual. Specifically, we combine an endpoint fidelity term with a Cauchy negative log-likelihood penalty (see App. A.3.5) under the induced bridge:

$$\mathcal{L}_{\text{end}} = \|\hat{x}_0 - x_0\|_2^2, \quad (11)$$

$$\mathcal{L}_{\text{cauchy}} = \sum_{i=1}^{d} \log\left( (x_{t,i} - \hat{\mu}_{t,i})^2 + \gamma_{t,i}^2 \right), \quad (12)$$

where constant terms independent of $\theta$ are omitted. Intuitively, $\mathcal{L}_{\text{cauchy}}$ encourages the simulated residual $x_t - \hat{\mu}_t$ to follow the prescribed heavy-tailed law, improving robustness to impulsive corruptions. Moreover, the Cauchy score (as a function of the residual) is *bounded and redescending*, which directly bounds the drift term used in reverse-time updates. On the learning side, it also yields a bounded supervision target for score matching, preventing extreme residuals from dominating the objective and improving numerical stability during training.

**Task-aligned spectrogram losses.** For bowel sound denoising, we represent $x_0$ in the spectrogram domain and optimize a task-aligned $\ell_1$ reconstruction loss together with a structural Gram loss to preserve high-level spectro-temporal patterns:

$$\mathcal{L}_{\text{rec}} = \|\hat{x}_0 - x_0\|_1, \quad (13)$$

$$\mathcal{L}_{\text{struct}} = \left\| G(\Phi(\hat{x}_0)) - G(\Phi(x_0)) \right\|_1, \quad (14)$$

where $\Phi(\cdot)$ is a fixed pretrained feature extractor (e.g., a VGG-style network on spectrograms) and $G(\cdot)$ is the Gram operator.

**Final objective.** Our final training objective combines task fidelity and heavy-tailed bridge consistency:

$$\mathcal{L}(\theta) = \lambda_{\text{rec}} \mathcal{L}_{\text{rec}} + \lambda_{\text{struct}} \mathcal{L}_{\text{struct}} + \lambda_{\text{c}} \mathcal{L}_{\text{cauchy}} \quad (15)$$

where the coefficients $\lambda_{\text{rec}}$, $\lambda_{\text{struct}}$ and $\lambda_{\text{c}}$ are hyperparameters that control the relative contributions of each loss term.

### 4.3. Cauchy-DBIM via Gaussian scale-mixture reparameterization

DBIM exploits the Gaussian bridge structure to derive an efficient closed-form backward update: given $x_{t_{n+1}}$ and the model prediction $\hat{x}_0$, one can compute $x_{t_n}$ in closed form (with optional Gaussian injection) without solving an SDE. This relies critically on finite-variance Gaussian noise and the tractability of Gaussian conditionals. For Cauchy-driven bridges, the noise has no finite moments, hence the original Gaussian DBIM update cannot be applied naively. We reparameterize the Cauchy driver as a Gaussian with a random scale (Gaussian scale mixture). For each coordinate, a standard Cauchy random variable can be generated as

$$\xi_i \sim \text{Cauchy}(0,1) \quad \Longleftrightarrow \quad \xi_i = \frac{z_i}{|u_i| + \varepsilon_u}, \quad (16)$$

where $z_i, u_i \overset{\text{i.i.d.}}{\sim} \mathcal{N}(0,1)$, and $\varepsilon_u > 0$ is a small constant for numerical stability ($\epsilon_u = 1e^{-10}$ by default). Define the random magnification factor

$$\kappa_i \triangleq \frac{s_i}{|u_i| + \varepsilon_u}, \qquad u_i \sim \mathcal{N}(0,1), \quad (17)$$

so that a Cauchy-like heavy-tailed residual with scale $s_i$ can be viewed as a Gaussian noise $z_i$ scaled by $\kappa_i$. Recall the bridge parameterization $\mu_t = a_t y + b_t x_0$ and the induced mean $\hat{\mu}_t = a_t y + b_t \hat{x}_0$ with $\hat{x}_0 = f_\theta(x_t, t, y)$. We define an *effective* (random) scale at time $t$:

$$\hat{\sigma}_{t,i} \triangleq \kappa_i \sigma_t, \qquad \hat{c}_{t,i} \triangleq \kappa_i c_t, \quad (18)$$

so conditional on $u$, the bridge residual behaves as Gaussian with variance $\hat{c}_{t,i}^2$. Importantly, to preserve the DBIM update structure under this reparameterization, the stochasticity parameter $\rho_n$ must be scaled by the same factor:

$$\hat{\rho}_{n,i} = \kappa_i \rho_n \quad \left(\text{equivalently, } \hat{\rho}_n = \kappa \odot \rho_n\right). \quad (19)$$

For the standard DBIM choice $\rho_n = \sigma_{t_n} \sqrt{1 - \frac{\text{SNR}_{t_{n+1}}}{\text{SNR}_{t_n}}}$ where $\alpha_t$ and $\sigma_t$ are pre-defined signal and noise schedules and $\text{SNR}_t = \frac{\alpha_t^2}{\sigma_t^2}$ denotes the signal-to-noise ratio at time $t$, if $\kappa$ is *shared across time* (i.e., the same $u$ is reused along the reverse trajectory), the SNR ratio remains unchanged and the scaling in (19) follows directly.

With the effective coefficients $\hat{c}_t = (\hat{c}_{t,i})_{i=1}^d$ and $\hat{\rho}_n = (\hat{\rho}_{n,i})_{i=1}^d$, we retain the Gaussian DBIM refinement rule and simply substitute $(c_t, \rho_n) \mapsto (\hat{c}_t, \hat{\rho}_n)$. Consider a time discretization $T = t_N > \cdots > t_0 = 0$. At step $n$, given the current state $x_{t_{n+1}}$, we first predict the clean endpoint

$$\hat{x}_0 = f_\theta(x_{t_{n+1}}, t_{n+1}, y),$$

which determines the corresponding bridge mean $\hat{\mu}_{t_n}$ and $\hat{\mu}_{t_{n+1}}$ (cf. Eq. (8)). We then update $x_{t_{n+1}} \mapsto x_{t_n}$ via

$$x_{t_n} = \hat{\mu}_{t_n} + \sqrt{\hat{c}_{t_n}^2 - \hat{\rho}_n^2} \odot \frac{x_{t_{n+1}} - \hat{\mu}_{t_{n+1}}}{\hat{c}_{t_{n+1}}} + \hat{\rho}_n \odot z_n, \quad (20)$$

where $z_n \sim \mathcal{N}(0, I)$ and all operations are element-wise. The deterministic (probability-flow) limit is recovered by setting $\rho_n = 0$ (hence $\hat{\rho}_n = 0$). We provide the derivation of the Gaussian scale-mixture reparameterization, together with the validity of the resulting Cauchy-DBIM refinement update, in App. A.4. Importantly, this refinement inherits the bounded and redescending influence induced by the Cauchy score (App. A.3.5), which prevents rare but extreme impulsive deviations from causing arbitrarily large drifts during sampling. The complete sampling procedure is summarized in Alg. 1 in App. A.4.

## 5. Experiments

### 5.1. Experimental Setup

#### 5.1.1. DATASETS

The CLINBS dataset, a large-scale collection of clinical bowel sounds (BS), serves as a robust resource for the development and evaluation of models capable of identifying gastrointestinal acoustic patterns under real-world conditions. The dataset was curated over a six-month period at the Gastroenterology Department, encompassing a diverse cohort of 1531 patients, with 76.79% of the samples categorized as normal bowel sounds and 23.21% as abnormal, including cases of gas-liquid gurgles (18.93%) and cases of metallic transients (4.29%). Recordings were captured using a custom-modified stethoscope integrated with a microphone, ensuring high-fidelity data acquisition in a quiet ward environment. Each sample is approximately one minute in duration; the dataset's annotations were rigorously cross-verified by multiple gastroenterologists to ensure high label integrity. For more detailed breakdowns of the dataset, including patient demographics and the distribution of specific pathological conditions, please refer to App. B.1.

To model speech interference, we use speech recordings as noise and conduct experiments on two standard corpora, VCTK (Yamagishi et al., 2019) and AISHELL (Fu et al., 2021), with details provided in App. B.2.

#### 5.1.2. EVALUATION METRICS

We evaluate the denoising performance across four dimensions to ensure a comprehensive assessment.

**Numerical Reconstruction:** **Mean Absolute Error (MAE)** and **Peak Signal-to-Noise Ratio (PSNR)** are used to quantify amplitude consistency and resilience to impulsive noise, while the **Pearson Correlation Coefficient**

*Table 1.* Quantitative comparison with state-of-the-art baselines.

| Model | Numerical | | | Structural | | Auditory | Distribution |
|---|---|---|---|---|---|---|---|
| | MAE $\downarrow$ | PSNR $\uparrow$ | PCC $\uparrow$ | SSIM $\uparrow$ | LPIPS $\downarrow$ | LSD $\downarrow$ | FID $\downarrow$ |
| Demucs (Rouard et al., 2023) | 19.17 | 18.84 | 0.76 | 0.51 | 0.35 | 29.07 | 69.59 |
| Mel-Roformer (Wang et al., 2024) | 24.95 | 16.89 | 0.66 | 0.39 | 0.45 | 36.11 | 126.09 |
| SCNet (Tong et al., 2024) | 38.52 | 14.75 | 0.61 | 0.28 | 0.49 | 46.13 | 103.36 |
| BDBM (Kieu et al., 2025) | 34.38 | 13.32 | 0.66 | 0.24 | 0.49 | 48.92 | 69.23 |
| DDBM (Zhou et al., 2024) | 19.47 | 17.17 | 0.68 | 0.40 | 0.35 | 31.98 | 87.33 |
| DBIM (Zheng et al., 2025) | 16.49 | 18.23 | 0.77 | 0.57 | 0.29 | 25.21 | 42.71 |
| **Ours** | **8.28** | **24.39** | **0.93** | **0.72** | **0.16** | **14.03** | **36.99** |

**(PCC)** measures the preservation of temporal envelopes and rhythmic integrity.

**Structural Integrity: Structural Similarity Index Measure (SSIM)** evaluates the maintenance of low-frequency harmonic patterns in spectrograms. To prevent over-smoothing, we employ **Learned Perceptual Image Patch Similarity (LPIPS)** to assess deep-feature similarity, ensuring the retention of natural, stochastic textures.

**Auditory Fidelity: Logarithmic Spectral Distance (LSD)** is used to measure timbral distortion and spectral envelope alignment.

**Perceptual Distribution Consistency: Fréchet Inception Distance (FID)** measures the alignment between denoised and clean signal distributions in deep feature space, while **Inception Score (IS)** reflects sample clarity and variability.

### 5.1.3. BASELINES

We benchmark our proposed method against a comprehensive set of SOTA methods.

**Discriminative Baselines:** We select Demucs(V4) (Rouard et al., 2023), Mel-RoFormer (Wang et al., 2024), and SCNet (Tong et al., 2024) as representative baselines. These models utilize Convolutional, Transformer, and Hybrid architectures, respectively, covering the dominant encoder–decoder frameworks in current literature.

**Generative Baseline:** We compare against BDBM (Kieu et al., 2025), a diffusion-based model that employs bidirectional denoising. This allows for a direct evaluation of our method against existing generative strategies.

All baselines were reproduced using official codebases and default configurations to ensure a fair comparison. All models used standardized 16kHz Mel-spectrograms (128 bands, FFT 1024, hop 256). Demucs (V4): HTDemucs architecture (4 stages, chunk size 343,980), Adam (lr=3e-4), 360 epochs. Mel-RoFormer: Frequency-domain Transformer (12 layers, 8 heads, dim 384), Adam (lr=1e-4), EMA 0.999.SCNet: Band-split sparse compression (4 subbands), Adam (lr=1e-4), multi-scale spectral loss. BDBM/DDBM/DBIM: All use

the official UNet backbone and 3000 training iterations. For inference, BDBM/DDBM/DBIM used 20 steps (matching our setting) with default schedules (VP-SDE for DDBM; original $\alpha_t, \sigma_t$ for DBIM).

### 5.1.4. IMPLEMENTATION DETAILS

**Experimental Setup** All models were implemented in PyTorch and trained on NVIDIA 4090 GPUs. The input features were single-channel Mel spectrograms extracted from 5-second audio clips at a sampling rate of 16 kHz. Spectrograms were computed using the following STFT parameters: FFT size of 1024, hop length of 256, window length of 1024, and 128 Mel bands. Prior to being fed into the models, the amplitudes of each Mel band were normalized to zero mean and unit variance.

**Network Architecture** Our generative model is based on a UNet architecture with a base channel size of 128. The number of channels is doubled at each resolution level (1, 2, 4, 8), with four residual blocks at each level. Attention layers are applied at intermediate downsampling scales, and each residual block incorporates temporal step embeddings to condition the model on the diffusion time step.

**Training Settings** Training is performed using the Adam optimizer with a base learning rate of $1 \times 10^{-4}$ and no weight decay. The global batch size is set to 256, with gradient accumulation implemented via a micro-batch size of 8. Exponential moving averages are maintained during training with a decay rate of 0.9999. Model checkpoints are saved every 10,000 iterations, and validation is performed every 500 iterations.

**Scheduling and bridge coefficients** We follow DBIM (Zheng et al., 2025) for all schedule-related components. In particular, we reuse the same base diffusion schedule parameters $(\alpha_t, \sigma_t)$ and the corresponding closed-form bridge parameterization induced by DBIM, namely $(a_t, b_t, c_t)$. As a result, our Cauchy-bridge instantiation preserves the DBIM mean path $\mu_t = a_t y + b_t x_0$ and the scale schedule $c_t$, while only replacing the Gaussian residual distribution with the proposed heavy-tailed noise model. For completeness, the exact definitions of

*Table 2.* Ablation studies on the framework. **Bold** indicates the best performance, and underlined indicates the second best.

| Model | PSNR↑ | SSIM↑ | LPIPS↓ | PCC↑ | MAE↓ | LSD↓ |
|---|---|---|---|---|---|---|
| Gaussian (DBIM) | 18.23 | 0.571 | 0.296 | 0.778 | 16.49 | 25.21 |
| Gaussian_l1 (DBIM) | 21.94 | 0.585 | 0.253 | 0.794 | 12.47 | 19.71 |
| Gaussian_gram (DBIM) | 20.03 | 0.602 | 0.218 | 0.817 | 14.12 | 22.22 |
| Cauchy (DBIM) | 20.08 | 0.594 | 0.292 | 0.802 | 13.25 | 18.14 |
| Cauchy_l1 (DBIM) | 22.11 | 0.608 | 0.269 | 0.835 | 10.90 | 14.69 |
| Cauchy_gram (DBIM) | 21.24 | 0.619 | 0.251 | 0.829 | 11.19 | 14.81 |
| Gaussian_l1_gram (DBIM) | 22.55 | 0.627 | 0.207 | 0.875 | 10.52 | 14.54 |
| Gaussian_l1_gram (GSM) | 22.67 | 0.630 | 0.201 | 0.880 | 10.36 | 14.38 |
| Cauchy_l1_gram (DBIM) | 23.21 | 0.706 | 0.191 | 0.883 | 9.54 | 14.12 |
| **Cauchy_l1_gram (GSM, Ours)** | **24.39** | **0.720** | **0.160** | **0.930** | **8.28** | **14.03** |

these schedules are deferred to App. A.2. To guarantee reproducibility, a fixed random seed was used during training and evaluation.

## 5.2. Main Results

We evaluate our framework against SOTA baselines, including source separation models (Demucs, Mel-Roformer, SCNet) and generative diffusion-based approaches (BDBM, DDBM, DBIM). As shown in Tab. 1, our method establishes the SOTA by outperforming all competitors across numerical, structural, and auditory metrics. Notably, our method achieves a MAE of 8.28, nearly doubling the reconstruction precision of the baseline. The high PSNR (24.39), PCC (0.93) and SSIM (0.72) further underscore the efficacy of the Cauchy-driven bridge in recovering precise signal amplitudes and spectro-temporal structures from heavy impulsive interference. Furthermore, our approach yields significantly better auditory and distributional results, achieving an LSD of 14.03 and an FID of 36.99. These gains suggest that the Cauchy prior is inherently better suited to the heavy-tailed nature of speech interference than standard Gaussian priors. Collectively, the improvements validate the robustness of our method for preserving diagnostic features in clinical settings, visual comparisons of the mel-spectrograms across different methods and additional examples of mel-spectrogram visualizations generated by our framework are provided in App. C.1.

## 5.3. Ablation Studies

We conducted ablation studies to quantify the individual contributions of components in Tab. 2. The Full Model demonstrates superior performance with SOTA scores in MAE (8.28), PCC (0.930), and SSIM (0.720). The inclusion of Gram Loss enhances structural fidelity and time-frequency alignment, yielding leading SSIM scores; while the synergy between the Cauchy cascade and $L_1$ loss proves essential for signal stability, as the heavy-tailed Cauchy prior effectively captures non-stationary acoustic events while the deterministic $L_1$ term ensures precise signal anchoring at the trajectory

---

**Algorithm 1** Cauchy-DBIM Sampling with $\rho_n$ Scaling (Gaussian Scale Mixture)

---

**Require:** Observation $y$; time grid $\{t_n\}_{n=0}^N$; schedules $\{a_t, b_t, c_t, \sigma_t, \mathrm{SNR}_t\}$; per-dim scales $\{s_i\}_{i=1}^d$; model $f_\theta$; $\varepsilon_u > 0$; clamp margin $\delta \in (0, 1)$.

**Ensure:** Sample $\tilde{x}_0$.

1: Initialize $x_{t_N} \leftarrow y$
2: Sample a **shared** $u \sim \mathcal{N}(0, I)$ and set $\kappa_i \leftarrow \frac{s_i}{|u_i| + \varepsilon_u}$ for all $i$
3: **for** $n = N-1, \ldots, 0$ **do**
4:     $\hat{x}_0 \leftarrow f_\theta(x_{t_{n+1}}, t_{n+1}, y)$
5:     $\hat{\mu}_{t_n} \leftarrow a_{t_n} y + b_{t_n} \hat{x}_0$;   $\hat{\mu}_{t_{n+1}} \leftarrow a_{t_{n+1}} y + b_{t_{n+1}} \hat{x}_0$
6:     Set $\hat{c}_{t_n,i} \leftarrow \kappa_i c_{t_n}$ and $\hat{c}_{t_{n+1},i} \leftarrow \kappa_i c_{t_{n+1}}$ for all $i$
7:     {DBIM base stochasticity (Gaussian)}
8:     $\rho_n \leftarrow \sigma_{t_n} \sqrt{1 - \mathrm{SNR}_{t_{n+1}}/\mathrm{SNR}_{t_n}}$
9:     {Scale $\rho_n$ by the same heavy-tail factor}
10:    Set $\hat{\rho}_{n,i} \leftarrow \kappa_i \rho_n$ for all $i$
11:    Clamp element-wise: $\hat{\rho}_{n,i} \leftarrow \min(\hat{\rho}_{n,i}, (1 - \delta)\hat{c}_{t_n,i})$ for all $i$
12:    Sample $z_n \sim \mathcal{N}(0, I)$
13:    $x_{t_n} \leftarrow \hat{\mu}_{t_n} + \sqrt{\hat{c}_{t_n}^2 - \hat{\rho}_n^2} \odot \frac{x_{t_{n+1}} - \hat{\mu}_{t_{n+1}}}{\hat{c}_{t_{n+1}}} + \hat{\rho}_n \odot z_n$
14: **end for**
15: **return** $\tilde{x}_0 \leftarrow x_{t_0}$

---

endpoint. Consequently, removing these components leads to a substantial performance drop, highlighting the necessity of our robust bridge formulation for preserving diagnostic bowel sound features under complex speech interference.

## 5.4. Shared vs. Resampled $u$ in Cauchy-DBIM Sampling

Since fixing $u$ across reverse steps (Alg. 1) makes per-step residuals conditionally Gaussian rather than exactly Cauchy, we emphasize that the exact Cauchy property pertains to the *forward bridge kernel* during training (Proposition 4.1). The Cauchy-DBIM sampler is a GSM-based inference procedure and does not require each conditional step to be exactly Cauchy. Sharing $u$ is a variance-reduction strat-

*Table 3.* Shared-$u$ vs. per-step resampled $u$ in Cauchy-DBIM.

| Strategy | MAE ↓ | SSIM ↑ | LSD ↓ |
|---|---|---|---|
| **Shared $u$ (ours)** | **8.28** | **0.72** | **14.03** |
| Resampled $u$ | 8.75 | 0.70 | 14.84 |

*Table 4.* Effect of Denoising on Bowel Sound Recognition.

| Method | ConvNeXt | Transformer | ResNet |
|---|---|---|---|
| Noised | 48.95% | 47.95% | 48.23% |
| Demucs | 52.00% | 62.38% | 66.23% |
| SCNet | 61.00% | 65.05% | 65.45% |
| Roformer | 52.00% | 53.10% | 62.87% |
| BDBM | 57.29% | 61.23% | 68.34% |
| DDBM | 78.16% | 82.51% | 82.18% |
| DBIM | 80.83% | 84.11% | 84.39% |
| **Ours** | **81.25%** | **87.45%** | **88.01%** |

*Table 5.* Quantitative results on the HLS-CMDS dataset.

| Metric | HS | LS | Mix |
|---|---|---|---|
| PSNR ↑ | 23.44 | 23.12 | 22.17 |
| SSIM ↑ | 0.74 | 0.67 | 0.76 |
| LPIPS ↓ | 0.20 | 0.26 | 0.18 |
| LSD ↓ | 12.53 | 14.33 | 14.22 |
| PCC ↑ | 0.81 | 0.79 | 0.80 |
| MAE ↓ | 11.71 | 12.73 | 12.25 |

*Table 6.* Performance with and without additional Gaussian noise on top of speech interference.

| Metric | Speech only | Speech + Gaussian |
|---|---|---|
| PSNR ↑ | 24.39 | 23.51 |
| SSIM ↑ | 0.72 | 0.70 |
| LPIPS ↓ | 0.16 | 0.18 |
| PCC ↑ | 0.93 | 0.91 |
| MAE ↓ | 8.28 | 9.10 |
| LSD ↓ | 14.03 | 13.95 |

egy: with fixed $u$, the effective scale $\hat{c}_{t,i} = \kappa_i c_t$ is consistent across steps, stabilizing the reverse trajectory and enabling closed-form DBIM updates. The per-step *marginal* remains heavy-tailed, as marginalizing out $u$ recovers a (regularized) Cauchy distribution with scale $c_{t_n} s_i$ (App. A.4), preserving the bounded-influence property of the Cauchy score (App. A.3.5). Empirical comparison (Tab. 3) shows that shared $u$ slightly outperforms per-step resampling, confirming that fixing the effective scale improves trajectory stability without compromising heavy-tailed robustness.

### 5.5. Bowel Sound Recognition

To evaluate the proposed framework in realistic clinical scenarios, we assess denoising performance via abnormal bowel sound recognition. Clean recordings are mixed with non-stationary speech interference from the AISHELL dataset ($\lambda = 0.5$) to simulate ward environments. We compare our method against SOTA approaches, including Demucs (V4), Mel-RoFormer, SCNet, and BDBM. As shown in Tab. 4, our method achieves the highest accuracy across all architectures. These results demonstrate that the framework effectively suppresses interference while preserving diagnostic acoustic features. By improving the abnormal bowel sound recognition performance, the system can support clinical diagnosis of gastrointestinal auscultation.

### 5.6. Evaluation Across Other Human Acoustic Signals

To further validate the generalisability of the proposed denoising framework, we conducted experiments on the HLS-CMDS dataset (Torabi et al., 2025). This dataset encompasses various human physiological acoustic signals, including heart sounds (HS), lung sounds (LS), and mixed cardiopulmonary sounds (MIX). We introduced speech interference from the AISHELL dataset and blended it with pure physiological signals using a mixing coefficient $\lambda = 0.5$

to emulate the speech interference. The results in Tab. 5 show that our method can effectively adapt to the influence of human speech on various physiological acoustic signals.

### 5.7. Robustness to Additional Gaussian Noise

In clinical environments, recordings may contain Gaussian-distributed noise (e.g., electronic sensor noise, ambient hum) in addition to speech interference. To evaluate robustness under such mixed-noise conditions, we added Gaussian noise ($\sigma = 0.01$) on top of the standard speech interference setup. As shown in Tab. 6, performance degrades only modestly, with LSD remaining essentially unchanged, indicating that spectral fidelity is well preserved. This suggests that, although our framework is primarily designed for heavy-tailed speech interference, it remains reasonably robust when additional Gaussian noise is present.

## 6. Conclusion

In this paper, we propose a novel generative framework for bowel sound denoising and clinical gastrointestinal assessment. We introduce CLINBS, a large-scale clinical dataset encompassing a diverse range of expert-verified pathological sounds. By formulating a Cauchy-driven Diffusion Bridge, we address the fundamental limitations of traditional Gaussian-based models in capturing the impulsive, heavy-tailed characteristics of speech interference. Extensive evaluations demonstrate that our framework achieves SOTA performance across all denoising metrics and significantly enhances the reliability of abnormal bowel sound recognition, providing a robust, non-invasive tool.

## Acknowledgement

This work is supported in part by Natural Science Foundation of Shanghai (No. 24ZR1430600), National Key Research and Development Program (No. 2025YFE0205200) and Shanghai Key Laboratory of Trusted Data Circulation and Governance, and Web3.

## Impact Statement

This research significantly advances non-invasive gastrointestinal monitoring by enabling high-fidelity bowel sound restoration in real-world clinical settings. By overcoming the long-standing challenge of speech interference through a novel generative framework, our work provides clinicians with a robust, longitudinal tool for early diagnosis and continuous management of digestive disorders. Furthermore, the introduction of the large-scale CLINBS dataset fosters the development of more equitable and accurate diagnostic AI, potentially reducing the reliance on resource-intensive and invasive procedures while improving patient outcomes across diverse clinical environments.

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

## A. Appendix: Cauchy-Driven Diffusion Bridge Models

### A.1. Diffusion Models

Given a data distribution $q_0(x_0)$ over $x_0 \in \mathbb{R}^d$, diffusion models construct a continuous-time stochastic process by defining a forward stochastic differential equation (SDE) that progressively transforms data into noise (Sohl-Dickstein et al., 2015; Ho et al., 2020; Song et al., 2021):

$$\mathrm{d}x_t = f(x_t, t)\mathrm{d}t + g(t)\mathrm{d}w_t \tag{21}$$

where $w_t$ is a standard Wiener process, $f : [0, T] \to \mathbb{R}$ is the drift coefficient, and $g : [0, T] \to \mathbb{R}$ controls the diffusion strength. When the forward SDE is linear with additive Gaussian noise, it admits an analytic transition kernel. In particular, for appropriately chosen noise schedules $\alpha_t$ and $\sigma_t$, the marginal distribution of $x_t$ conditioned on $x_0$ (Itô, 1951):

$$q(x_t \mid x_0) = \mathcal{N}\big(\alpha_t x_0, \sigma_t^2 I\big), \tag{22}$$

where $\alpha_t \in (0, 1]$ and $\sigma_t > 0$ satisfy (Kingma et al., 2021)

$$f(x_t, t) = \frac{d\log\alpha_t}{dt} x_t, \quad g(t)^2 = \frac{d\sigma_t^2}{dt} - 2\sigma_t^2 \frac{d\log\alpha_t}{dt} \tag{23}$$

The forward process induces a family of marginal distributions $\{q(x_t)\}_{t=0}^T$. By design, as $t \to T$, the signal component vanishes ($\alpha_t \to 0$) and the terminal distribution approaches a simple Gaussian prior: $q(x_T) \approx \mathcal{N}(0, \sigma_T^2 I)$.

Sampling from the data distribution can be achieved by reversing the diffusion process. The time-reversed SDE is given by (Song et al., 2021):

$$dx_t = \big[f(x_t, t) - g^2(t)\nabla_{x_t} \log q(x_t)\big] dt + g(t)\, d\bar{w}_t, \tag{24}$$

where $d\bar{w}_t$ denotes a reverse-time Wiener process and $\nabla_{x_t} \log q(x_t)$ is the score function of the marginal density at time $t$. Alternatively, one can define a deterministic process known as the probability flow ODE, which shares the same marginals as the reverse SDE:

$$dx_t = \left[f(x_t, t) - \frac{1}{2}g^2(t)\nabla_{x_t} \log q(x_t)\right] dt. \tag{25}$$

Since the true score function $\nabla_{x_t} \log q(x_t)$ is intractable, it is approximated by a neural network $s_\theta(x_t, t)$, trained via denoising score matching (DSM)(Vincent, 2011):

$$\min_\theta \mathbb{E}_{x_0 \sim q_0(x_0)} \mathbb{E}_{x_t \sim q(\cdot|x_0)} \left[w(t) \|s_\theta(x_t, t) - \nabla_{x_t} \log q(x_t \mid x_0)\|^2\right], \tag{26}$$

where $w(t)$ is a time-dependent weighting function. Once trained, the learned score model $s_\theta$ can be plugged into either the reverse SDE or the probability flow ODE to generate samples from $q_0(x_0)$.

### A.2. Denoising Diffusion Bridge Models

For paired data $(x_0, y) \sim p_{\text{data}}(x_0, y)$, denoising diffusion bridge models (DDBM) (Zhou et al., 2024) extend standard diffusion models by conditioning the stochastic process to arrive at the prescribed endpoint $y \in \mathbb{R}^d$. This is achieved by applying Doob's $h$-transform (Doob & Doob, 1984) to the forward diffusion SDE in Eq. 21, resulting in a conditioned stochastic process that forms a diffusion bridge:

$$dx_t = f(x_t, t)\, dt + g^2(t)\nabla_{x_t} \log q(x_T = y \mid x_t)\, dt + g(t)\, dw_t, \quad (x_0, y) \sim p_{data}(x, y),\ x_T = y. \tag{27}$$

Unlike standard diffusion models, where the endpoint is sampled from a Gaussian prior, the conditioning variable $y$ in diffusion bridges can be an arbitrary informative target, such as a degraded observation in image restoration or inverse problems. For Gaussian diffusion processes, the bridge process in Eq. 27 admits an analytic forward transition kernel. In particular, conditioned on both $x_0$ and $x_T = y$, the intermediate state follows:

$$q(x_t \mid x_0, x_T = y) = \mathcal{N}(a_t y + b_t x_0, c_t^2 I), \tag{28}$$

where

$$a_t = \frac{\alpha_t}{\alpha_T} \frac{\text{SNR}_T}{\text{SNR}_t}, \quad b_t = \alpha_t \left(1 - \frac{\text{SNR}_T}{\text{SNR}_t}\right), \quad c_t^2 = \sigma_t^2 \left(1 - \frac{\text{SNR}_T}{\text{SNR}_t}\right), and \quad \text{SNR}_t = \frac{\alpha_t^2}{\sigma_t^2} \tag{29}$$

$\alpha_t$ and $\sigma_t$ are pre-defined signal and noise schedules and $\text{SNR}_t$ denotes the signal-to-noise ratio at time $t$. This distribution explicitly interpolates between the data sample $x_0$ and the endpoint constraint $y$, forming a Gaussian diffusion bridge. Similar to standard diffusion models, the diffusion bridge process admits a reverse-time SDE (Song et al., 2021):

$$dx_t = \left[f(x_t, t) - g^2(t)\left(\nabla_{x_t} \log q(x_t \mid x_T = y) - \nabla_{x_t} \log q(x_T = y \mid x_t)\right)\right] dt + g(t)\, d\bar{w}_t, \tag{30}$$

where $\bar{w}_t$ is the reverse-time Wiener process and $\nabla_{x_t} \log q(x_T = y \mid x_t)$ is analytically known for Gaussian diffusions. Equivalently, the corresponding probability flow ODE is given by

$$dx_t = \left[f(x_t, t) - g^2(t)\left(\frac{1}{2}\nabla_{x_t} \log q(x_t \mid x_T = y) - \nabla_{x_t} \log q(x_T = y \mid x_t)\right)\right] dt. \tag{31}$$

Both dynamics share the same marginal distributions $\{q(x_t \mid x_T = y)\}_{t=0}^{T}$ as the forward bridge process.

In Eqs. 30- 31, the only unknown quantity is the bridge score $\nabla_{x_t} \log q(x_t \mid x_T = y)$, which captures how the diffusion trajectory is attracted toward the endpoint constraint $y$. DDBM proposes to learn this score function using denoising bridge score matching (DBSM). Specifically, a neural network $s_\theta(x_t, t, y)$ is trained to minimize:

$$\mathcal{L}_{BSM}(\theta) = \mathbb{E}_t \mathbb{E}_{(x_0, y) \sim p_{\text{data}}(x_0, y)} \mathbb{E}_{x_t \sim q(x_t \mid x_0, x_T = y)} \left[w(t) \|s_\theta(x_t, t, y) - \nabla_{x_t} \log q(x_t \mid x_0, x_T = y)\|^2\right] \tag{32}$$

where $w(t)$ is a positive weighting function and $q(x_t \mid x_0, x_T = y)$ is given by Eq. 28. Once trained, the learned bridge score can be substituted into the reverse SDE or probability flow ODE to generate samples conditioned on $y$.

Although DDBM significantly extends diffusion models beyond unconditional generation, they fundamentally rely on Gaussian diffusion bridges, inheriting finite-variance noise and Brownian-driven dynamics. As a result, the induced bridge trajectories exhibit light-tailed behavior, which may be suboptimal when modeling heavy-tailed, impulsive, or sparse conditional distributions.

## A.3. Cauchy-Driven Diffusion Bridge Models

### A.3.1. MOTIVATION: HEAVY-TAILED CORRUPTION IN GI DENOISING

We study Bowel sound denoising under the observation model

$$y = x + n, \qquad x \sim p_{\text{clean}}, \ y \sim p_{\text{noisy}}, \tag{33}$$

where $x$ denotes a clean bowel sound signal, $y$ is the noisy observation, and $n$ mainly arises from speech and environmental interference. In practical recordings, such interference is often impulsive and heavy-tailed (e.g., bursty speech segments and non-stationary outliers), violating Gaussian residual assumptions. Since both Gaussian ($\alpha = 2$) and Cauchy ($\alpha = 1$) distributions are $\alpha$-stable, we replace the Gaussian noise driver in denoising diffusion bridge models with an i.i.d. Cauchy driver to better match the corruption statistics and improve robustness.

**Proposition 1.** Let $\alpha \in (0, 2]$. A real-valued random variable $X$ is called $\alpha$-stable (stable with stability index $\alpha$) if for any $a, b \in \mathbb{R}$, letting $X_1, X_2$ be i.i.d. copies of $X$, there exist constants $c > 0$ and $d \in \mathbb{R}$ such that

$$aX_1 + bX_2 \overset{d}{=} cX + d. \tag{34}$$

If one can always choose $d = 0$, $X$ is called strictly $\alpha$-stable, in which case the scale factor necessarily takes the form

$$c = (|a|^\alpha + |b|^\alpha)^{1/\alpha}. \tag{35}$$

Then:

1. If $X \sim \mathcal{N}(\mu, \sigma^2)$, then $X$ is $\alpha$-stable with $\alpha = 2$. Moreover, if $\mu = 0$, then $X$ is strictly 2-stable and

$$aX_1 + bX_2 \overset{d}{=} \sqrt{a^2 + b^2}\, X. \tag{36}$$

2. If $X \sim \mathrm{Cauchy}(x_0, \gamma)$, then $X$ is $\alpha$-stable with $\alpha = 1$. Moreover, if $x_0 = 0$, then $X$ is strictly 1-stable and

$$aX_1 + bX_2 \stackrel{d}{=} (|a| + |b|)\, X. \tag{37}$$

*Proof.* **Gaussian case.** The characteristic function of $X \sim \mathcal{N}(\mu, \sigma^2)$ is

$$\varphi_X(t) = \mathbb{E}[e^{itX}] = \exp\!\Big(i\mu t - \tfrac{1}{2}\sigma^2 t^2\Big). \tag{38}$$

For i.i.d. $X_1, X_2$ and any $a, b \in \mathbb{R}$,

$$\varphi_{aX_1+bX_2}(t) = \varphi_X(at)\, \varphi_X(bt) = \exp\!\Big(i\mu(a+b)t - \tfrac{1}{2}\sigma^2(a^2+b^2)t^2\Big),$$

which is the characteristic function of $\mathcal{N}\big((a+b)\mu,\ (a^2+b^2)\sigma^2\big)$. Let $c = \sqrt{a^2 + b^2}$ and $d = (a + b - c)\mu$. Since

$$cX + d \sim \mathcal{N}(c\mu + d,\ c^2\sigma^2) = \mathcal{N}\big((a+b)\mu,\ (a^2+b^2)\sigma^2\big), \tag{39}$$

we obtain $aX_1 + bX_2 \stackrel{d}{=} cX + d$, establishing stability. If $\mu = 0$, then $d = 0$ and the strict form holds with $c = (|a|^2 + |b|^2)^{1/2}$, hence $\alpha = 2$.

**Cauchy case.** The characteristic function of $X \sim \mathrm{Cauchy}(x_0, \gamma)$ is

$$\varphi_X(t) = \mathbb{E}[e^{itX}] = \exp\!\big(itx_0 - \gamma|t|\big). \tag{40}$$

For i.i.d. $X_1, X_2$ and any $a, b \in \mathbb{R}$,

$$\varphi_{aX_1+bX_2}(t) = \varphi_X(at)\, \varphi_X(bt) = \exp\!\big(it(a+b)x_0 - \gamma(|a| + |b|)|t|\big),$$

which is the characteristic function of $\mathrm{Cauchy}\big((a+b)x_0,\ \gamma(|a| + |b|)\big)$. Using the affine invariance $cX + d \sim \mathrm{Cauchy}(cx_0 + d,\ |c|\gamma)$, choose

$$c = |a| + |b|, \qquad d = \big(a + b - (|a| + |b|)\big)x_0, \tag{41}$$

so that $cx_0 + d = (a + b)x_0$ and $|c|\gamma = \gamma(|a| + |b|)$, yielding $aX_1 + bX_2 \stackrel{d}{=} cX + d$. If $x_0 = 0$, then $d = 0$ and the strict form holds with $c = |a| + |b| = (|a|^1 + |b|^1)^1$, hence $\alpha = 1$. $\qquad\square$

### A.3.2. CAUCHY FORWARD BRIDGE KERNEL

Following denoising diffusion bridge models (DDBM), we adopt a linear conditional forward kernel that interpolates between the clean endpoint $x_0$ and the observation endpoint $x_T = y$:

$$x_t = a_t\, y + b_t\, x_0 + c_t\, \varepsilon, \qquad \varepsilon \sim \mathrm{Cauchy}(0, r)^{\otimes d}, \tag{42}$$

where $\varepsilon$ has i.i.d. coordinates drawn from a univariate Cauchy distribution with scale $r > 0$, and $a_t, b_t, c_t$ are time-dependent schedules. To ensure endpoint consistency, we impose the standard bridge constraints

$$a_0 = 0,\ b_0 = 1,\ c_0 = 0; \qquad a_T = 1,\ b_T = 0,\ c_T = 0, \tag{43}$$

so that $x_{t=0} = x_0$ and $x_{t=T} = y$ deterministically.

**Proposition 4.1** [Existence of the Cauchy bridge kernel] Fix a dimension $d \geq 1$. Let $(a_t, b_t, c_t)_{t \in [0,T]}$ be deterministic schedules such that $(a_0, b_0, c_0) = (0, 1, 0), (a_T, b_T, c_T) = (1, 0, 0)$, and $c_t > 0$ for all $t \in (0, T)$. Let $\varepsilon = (\varepsilon_1, \ldots, \varepsilon_d)$ have independent coordinates $\varepsilon_i \sim \mathrm{Cauchy}(0, r_i)$, where $r_i > 0$ (the i.i.d. case corresponds to $r_i \equiv r$). For any fixed endpoints $(x_0, y) \in \mathbb{R}^d \times \mathbb{R}^d$, define for each $t \in [0, T]$

$$x_t \stackrel{\triangle}{=} a_t y + b_t x_0 + c_t \varepsilon. \tag{44}$$

Then:

1. (Endpoint satisfaction.) One has $x_{t=0} = x_0$ and $x_{t=T} = y$ almost surely.

2. (Existence of a probability kernel.) For each $t \in [0, T]$, the conditional law

$$q_t(\cdot \mid x_0, y) \triangleq \mathbb{P}(x_t \in \cdot \mid x_0, y)$$

   defines a well-defined probability kernel on $(\mathbb{R}^d, \mathcal{B}(\mathbb{R}^d))$.

3. (Explicit density for $t \in (0, T)$.) For $t \in (0, T)$, $q_t(\cdot \mid x_0, y)$ is absolutely continuous with respect to Lebesgue measure, and admits the factorized density

$$q(x_t \mid x_0, y) = \prod_{i=1}^{d} \frac{1}{\pi} \frac{\gamma_{t,i}}{(x_{t,i} - \mu_{t,i})^2 + \gamma_{t,i}^2}, \qquad \mu_t(x_0, y) \triangleq a_t y + b_t x_0, \quad \gamma_{t,i} \triangleq c_t r_i. \tag{45}$$

   In particular, in the i.i.d. case $r_i \equiv r$, we have $\gamma_{t,i} \equiv \gamma_t = c_t r$.

4. (Degenerate densities at endpoints.) At $t = 0$ and $t = T$, the kernels are degenerate:

$$q_0(\cdot \mid x_0, y) = \delta_{x_0}(\cdot), \qquad q_T(\cdot \mid x_0, y) = \delta_y(\cdot),$$

   where $\delta_z$ denotes the Dirac measure at $z$.

*Proof.* **1) Endpoint satisfaction.** By the schedule conditions and Eq. 44,

$$x_0 = a_0 y + b_0 x_0 + c_0 \varepsilon = 0 \cdot y + 1 \cdot x_0 + 0 \cdot \varepsilon = x_0, \tag{46}$$

and similarly

$$x_T = a_T y + b_T x_0 + c_T \varepsilon = 1 \cdot y + 0 \cdot x_0 + 0 \cdot \varepsilon = y, \tag{47}$$

almost surely.

**2) Kernel existence.** For any fixed $(x_0, y)$ and each $t \in [0, T]$, $x_t$ is a measurable function of $\varepsilon$ via Eq. 44. Since the map $\varepsilon \mapsto x_t(x_0, y, \varepsilon)$ is Borel-measurable (indeed affine), the conditional law of $x_t$ given $(x_0, y)$ is the pushforward measure of the Cauchy law under this map. Hence, for every Borel set (Martin, 1975) $A \in \mathcal{B}(\mathbb{R}^d)$,

$$q_t(A \mid x_0, y) \triangleq \mathbb{P}(x_t \in A \mid x_0, y) \tag{48}$$

is well-defined, and $A \mapsto q_t(A \mid x_0, y)$ is a probability measure. Moreover, for each fixed $A$, the mapping $(x_0, y) \mapsto q_t(A \mid x_0, y)$ is measurable, so $q_t(\cdot \mid x_0, y)$ defines a probability kernel.

**3) Explicit density for $t \in (0, T)$.** Fix $t \in (0, T)$, so that $c_t > 0$. Define $\mu_t = \mu_t(x_0, y) = a_t y + b_t x_0$. For each coordinate, we have the affine change of variables

$$x_{t,i} = \mu_{t,i} + c_t \varepsilon_i, \qquad i = 1, \ldots, d. \tag{49}$$

Since $\varepsilon_i \sim \text{Cauchy}(0, r_i)$, its density is

$$f_{\varepsilon_i}(u) = \frac{1}{\pi} \frac{r_i}{u^2 + r_i^2}. \tag{50}$$

By the one-dimensional change-of-variables formula with $u = (x_{t,i} - \mu_{t,i})/c_t$ and Jacobian $1/c_t$, the conditional density of $x_{t,i}$ given $(x_0, y)$ is

$$f_{x_{t,i} \mid x_0, y}(x) = \frac{1}{c_t} f_{\varepsilon_i}\left(\frac{x - \mu_{t,i}}{c_t}\right) = \frac{1}{c_t} \cdot \frac{1}{\pi} \frac{r_i}{\left(\frac{x - \mu_{t,i}}{c_t}\right)^2 + r_i^2} = \frac{1}{\pi} \frac{c_t r_i}{(x - \mu_{t,i})^2 + (c_t r_i)^2}. \tag{51}$$

Letting $\gamma_{t,i} \triangleq c_t r_i$ yields exactly the $\text{Cauchy}(\mu_{t,i}, \gamma_{t,i})$ density. Because the coordinates $\{\varepsilon_i\}_{i=1}^{d}$ are independent and the map $x_t = \mu_t + c_t \varepsilon$ acts coordinate-wise with shared scalar $c_t$, the joint conditional density factorizes:

$$q(x_t \mid x_0, y) = \prod_{i=1}^{d} f_{x_{t,i} \mid x_0, y}(x_{t,i}) = \prod_{i=1}^{d} \frac{1}{\pi} \frac{\gamma_{t,i}}{(x_{t,i} - \mu_{t,i})^2 + \gamma_{t,i}^2}, \tag{52}$$

Finally, each one-dimensional Cauchy density integrates to 1, hence the product integrates to 1 on $\mathbb{R}^d$, showing $q(\cdot \mid x_0, y)$ is a valid conditional density for $t \in (0, T)$.

**4) Endpoints are degenerate.** At $t = 0$ and $t = T$, we have $c_t = 0$ so $x_t$ becomes deterministic ($x_0$ or $y$), thus the conditional law is a Dirac measure as stated. $\qquad\square$

### A.3.3. CLOSED-FORM BRIDGE SCORE TARGET

A key advantage of Eq. 52 is that it provides an analytic *bridge score target* for denoising score matching. Taking the gradient of $\log q(x_t \mid x_0, y)$ w.r.t. $x_t$ yields a closed-form expression.

**Proposition 4.2** [Closed-form log-density and score of the Cauchy bridge kernel] Under the setting of Proposition 4.1, for any $t \in (0, T)$ the conditional density $q(x_t \mid x_0, y)$ is given by Eq. 45 in Appendix A.3.2 with $\mu_t = a_t y + b_t x_0$ and $\gamma_{t,i} = c_t r_i$. Moreover, its log-density and score (gradient w.r.t. $x_t$) admit closed forms:

$$
\log q(x_t \mid x_0, y) = -\sum_{i=1}^{d} \log\left(\pi \gamma_{t,i}\right) - \sum_{i=1}^{d} \log\left((x_{t,i} - \mu_{t,i})^2 + \gamma_{t,i}^2\right), \tag{53}
$$

$$
\nabla_{x_t} \log q(x_t \mid x_0, y) = \left(-\frac{2(x_{t,i} - \mu_{t,i})}{(x_{t,i} - \mu_{t,i})^2 + \gamma_{t,i}^2}\right)_{i=1}^{d}. \tag{54}
$$

*Proof.* Fix any $t \in (0, T)$ and endpoints $(x_0, y) \in \mathbb{R}^d \times \mathbb{R}^d$. Under Proposition 4.1, the conditional forward kernel admits the (coordinate-wise) Cauchy density

$$
q(x_t \mid x_0, y) = \prod_{i=1}^{d} \frac{1}{\pi} \frac{\gamma_{t,i}}{(x_{t,i} - \mu_{t,i})^2 + \gamma_{t,i}^2}, \qquad \mu_t \triangleq a_t y + b_t x_0, \quad \gamma_{t,i} \triangleq c_t r_i, \tag{55}
$$

where $\gamma_{t,i} > 0$ since $c_t > 0$ for $t \in (0, T)$ and $r_i > 0$ by definition.

**Closed-form log-density.** Taking logarithms on both sides of Eq. 55 yields

$$
\begin{aligned}
\log q(x_t \mid x_0, y) &= \sum_{i=1}^{d} \log\left(\frac{1}{\pi} \frac{\gamma_{t,i}}{(x_{t,i} - \mu_{t,i})^2 + \gamma_{t,i}^2}\right) \\
&= \sum_{i=1}^{d} \left[ -\log \pi + \log \gamma_{t,i} - \log\left((x_{t,i} - \mu_{t,i})^2 + \gamma_{t,i}^2\right) \right] \\
&= -\sum_{i=1}^{d} \log(\pi \gamma_{t,i}) - \sum_{i=1}^{d} \log\left((x_{t,i} - \mu_{t,i})^2 + \gamma_{t,i}^2\right). 
\end{aligned} \tag{56}
$$

This is exactly Eq. 53 in Proposition 4.2.

**Differentiability w.r.t. $x_t$.** For each coordinate $i$, define

$$
g_i(x_{t,i}) \triangleq (x_{t,i} - \mu_{t,i})^2 + \gamma_{t,i}^2.
$$

Since $\gamma_{t,i}^2 > 0$, we have $g_i(x_{t,i}) > 0$ for all $x_{t,i} \in \mathbb{R}$, hence $\log g_i(x_{t,i})$ is well-defined and smooth ($C^\infty$) in $x_{t,i}$. Therefore, $\log q(x_t \mid x_0, y)$ is differentiable w.r.t. $x_t$ and its gradient can be computed coordinate-wise.

**Closed-form score (gradient).** From Eq. 56, only the second summation depends on $x_t$. For each $i$,

$$
\begin{aligned}
\frac{\partial}{\partial x_{t,i}} \log q(x_t \mid x_0, y) &= -\frac{\partial}{\partial x_{t,i}} \log\big((x_{t,i} - \mu_{t,i})^2 + \gamma_{t,i}^2\big) \\
&= -\frac{1}{(x_{t,i} - \mu_{t,i})^2 + \gamma_{t,i}^2} \cdot \frac{\partial}{\partial x_{t,i}} \Big((x_{t,i} - \mu_{t,i})^2 + \gamma_{t,i}^2\Big) \\
&= -\frac{1}{(x_{t,i} - \mu_{t,i})^2 + \gamma_{t,i}^2} \cdot 2(x_{t,i} - \mu_{t,i}) \\
&= -\frac{2(x_{t,i} - \mu_{t,i})}{(x_{t,i} - \mu_{t,i})^2 + \gamma_{t,i}^2}.
\end{aligned}
\tag{57}
$$

Stacking the coordinates gives the vector-form score

$$
\nabla_{x_t} \log q(x_t \mid x_0, y) = \left( -\frac{2(x_{t,i} - \mu_{t,i})}{(x_{t,i} - \mu_{t,i})^2 + \gamma_{t,i}^2} \right)_{i=1}^{d},
\tag{58}
$$

which matches Eq. 54 in Proposition 4.2.

**Equivalent vector notation.** Let $\oslash$ denote elementwise division and $\odot$ elementwise multiplication, and let $\gamma_t^2 \triangleq (\gamma_{t,1}^2, \ldots, \gamma_{t,d}^2)$. Then Eq. 58 can be written compactly as

$$
\nabla_{x_t} \log q(x_t \mid x_0, y) = -2(x_t - \mu_t) \oslash \big((x_t - \mu_t)^{\odot 2} + \gamma_t^2\big),
\tag{59}
$$

This completes the proof. $\qquad\square$

### A.3.4. TRAINING OBJECTIVE.

Let $s_\theta(x_t, t, y)$ be a parameterized network for predicting the (conditional) bridge score. We minimize a denoising-score-matching objective analogous to DDBM/DBSM, replacing the Gaussian score target by Eq. 59:

$$
\mathcal{L}(\theta) = \mathbb{E}_{(x_0, y) \sim p_{\text{data}}(x_0, y)} \mathbb{E}_t \mathbb{E}_{x_t \sim q(\cdot \mid x_0, y)} \Big[ w(t) \big\| s_\theta(x_t, t, y) - \nabla_{x_t} \log q(x_t \mid x_0, y) \big\|_2^2 \Big],
\tag{60}
$$

where $w(t) > 0$ is a time weighting function.

**Proposition 4.3** [Endpoint regression as a plug-in bridge parameterization] Let $(X_0, Y)$ be a random pair with conditional density $p(x_0 \mid y)$, and let $q(x_t \mid x_0, y, t)$ be a bridge kernel such that for each $(x_0, y, t)$: (i) $q(\cdot \mid x_0, y, t)$ is a.e. positive and continuously differentiable in $x_t$; (ii) the marginal $q_t(x_t \mid y) := \int q(x_t \mid x_0, y, t) \, p(x_0 \mid y) \, dx_0$ is finite and positive a.e.; (iii) differentiation can be interchanged with integration, e.g., there exists an integrable dominating function $g(\cdot, y, t)$ such that $\|\nabla_{x_t} q(x_t \mid x_0, y, t)\| \le g(x_t, y, t)$ for $p(x_0 \mid y)$-a.e. $x_0$.

**(Fisher identity).** Then the marginal bridge score satisfies

$$
\nabla_{x_t} \log q_t(x_t \mid y) = \mathbb{E}[\nabla_{x_t} \log q(x_t \mid X_0, Y, t) \mid X_t = x_t, t, Y = y].
\tag{61}
$$

**($\ell_2$-optimal endpoint regression).** For any measurable predictor $f = f(x_t, t, y)$ with $\mathbb{E}\|X_0\|_2^2 < \infty$, the $\ell_2$ risk minimizer is the posterior mean:

$$
f^\star(x_t, t, y) = \arg\min_f \ \mathbb{E}\big[\|f(X_t, t, Y) - X_0\|_2^2\big] = \mathbb{E}[X_0 \mid X_t = x_t, t, Y = y].
\tag{62}
$$

**(Plug-in induced score and a concentration-based bound).** Assume moreover that the conditional score admits a closed form and depends on $x_0$ only through a bridge parameterization $\psi_t = \psi_t(x_0, y)$:

$$
\nabla_{x_t} \log q(x_t \mid x_0, y, t) = s(x_t, t, y; \psi_t(x_0, y)).
\tag{63}
$$

Suppose $s(x_t, t, y; \psi)$ is $L_s$-Lipschitz in $\psi$ for fixed $(x_t, t, y)$:

$$\|s(x_t, t, y; \psi_1) - s(x_t, t, y; \psi_2)\|_2 \le L_s \|\psi_1 - \psi_2\|_2. \tag{64}$$

Given any estimator $\hat{x}_0 = \hat{x}_0(x_t, t, y)$, define the *plug-in induced score*

$$s^{\text{ind}}(x_t, t, y) := s(x_t, t, y; \psi_t(\hat{x}_0, y)) = \nabla_{x_t} \log q(x_t \mid \hat{x}_0, y, t). \tag{65}$$

Then the induced score approximates the intractable marginal score with the bound

$$\left\| \nabla_{x_t} \log q_t(x_t \mid y) - s^{\text{ind}}(x_t, t, y) \right\|_2 \le \mathbb{E}\left[ \left\| s(x_t, t, y; \psi_t(X_0, y)) - s(x_t, t, y; \psi_t(\hat{x}_0, y)) \right\|_2 \,\middle|\, x_t, t, y \right]$$
$$\le L_s \, \mathbb{E}[\|\psi_t(X_0, y) - \psi_t(\hat{x}_0, y)\|_2 \mid x_t, t, y]. \tag{66}$$

In particular, if $\psi_t(\cdot, y)$ is $L_\psi$-Lipschitz in $x_0$, then

$$\left\| \nabla_{x_t} \log q_t(x_t \mid y) - s^{\text{ind}}(x_t, t, y) \right\|_2 \le L_s L_\psi \, \mathbb{E}[\|X_0 - \hat{x}_0\|_2 \mid x_t, t, y]. \tag{67}$$

Choosing $\hat{x}_0 = f^\star(x_t, t, y) = \mathbb{E}[X_0 \mid x_t, t, y]$ further gives

$$\mathbb{E}[\|X_0 - f^\star\|_2 \mid x_t, t, y] \le \sqrt{\operatorname{tr}(\operatorname{Cov}(X_0 \mid x_t, t, y))}, \tag{68}$$

so the approximation error becomes small whenever the posterior $p(x_0 \mid x_t, t, y)$ is concentrated.

*Proof.* We prove each claim.

**Fisher identity.** By definition,

$$q_t(x_t \mid y) = \int q(x_t \mid x_0, y, t) \, p(x_0 \mid y) \, dx_0. \tag{69}$$

Under assumption (iii), we can differentiate under the integral sign:

$$\nabla_{x_t} q_t(x_t \mid y) = \int \nabla_{x_t} q(x_t \mid x_0, y, t) \, p(x_0 \mid y) \, dx_0. \tag{70}$$

Divide both sides by $q_t(x_t \mid y)$ (assumption (ii)):

$$\nabla_{x_t} \log q_t(x_t \mid y) = \frac{1}{q_t(x_t \mid y)} \int \nabla_{x_t} q(x_t \mid x_0, y, t) \, p(x_0 \mid y) \, dx_0$$
$$= \int \nabla_{x_t} \log q(x_t \mid x_0, y, t) \, \frac{q(x_t \mid x_0, y, t) \, p(x_0 \mid y)}{q_t(x_t \mid y)} \, dx_0. \tag{71}$$

Recognize the fraction as the posterior density

$$p(x_0 \mid x_t, t, y) = \frac{q(x_t \mid x_0, y, t) \, p(x_0 \mid y)}{q_t(x_t \mid y)}. \tag{72}$$

Hence,

$$\nabla_{x_t} \log q_t(x_t \mid y) = \int \nabla_{x_t} \log q(x_t \mid x_0, y, t) \, p(x_0 \mid x_t, t, y) \, dx_0 = \mathbb{E}[\nabla_{x_t} \log q(x_t \mid X_0, Y, t) \mid x_t, t, y], \tag{73}$$

which is Eq. 61.

$\ell_2$ **Bayes estimator.** Fix $(x_t, t, y)$ and consider the conditional risk

$$R(f; x_t, t, y) := \mathbb{E}\left[ \|f(x_t, t, y) - X_0\|_2^2 \mid x_t, t, y \right]. \tag{74}$$

Expand the square and use $\mathbb{E}\|X_0\|_2^2 < \infty$:

$$R(f; x_t, t, y) = \|f(x_t, t, y)\|_2^2 - 2 \, f(x_t, t, y)^\top \mathbb{E}[X_0 \mid x_t, t, y] + \mathbb{E}[\|X_0\|_2^2 \mid x_t, t, y]. \tag{75}$$

The last term is independent of $f$; the first two terms form a strictly convex quadratic in $f$, minimized at $f^\star(x_t, t, y) = \mathbb{E}[X_0 \mid x_t, t, y]$, proving Eq. 62.

**Plug-in induced score bound.** By Eq. 63 and the Fisher identity Eq. 61,

$$\nabla_{x_t} \log q_t(x_t \mid y) = \mathbb{E}[s(x_t, t, y; \psi_t(X_0, y)) \mid x_t, t, y]. \tag{76}$$

Therefore,

$$
\begin{aligned}
\left\| \nabla_{x_t} \log q_t(x_t \mid y) - s^{\mathrm{ind}}(x_t, t, y) \right\|_2 &= \left\| \mathbb{E}[s(x_t, t, y; \psi_t(X_0, y)) \mid x_t, t, y] - s(x_t, t, y; \psi_t(\hat{x}_0, y)) \right\|_2 \\
&\leq \mathbb{E}\big[ \left\| s(x_t, t, y; \psi_t(X_0, y)) - s(x_t, t, y; \psi_t(\hat{x}_0, y)) \right\|_2 \,\big|\, x_t, t, y \big],
\end{aligned}
\tag{77}
$$

where we used Jensen's inequality for the norm. Applying the Lipschitz condition Eq. 64 yields Eq. 66. If $\psi_t(\cdot, y)$ is $L_\psi$-Lipschitz in $x_0$, then $\|\psi_t(X_0, y) - \psi_t(\hat{x}_0, y)\|_2 \leq L_\psi \|X_0 - \hat{x}_0\|_2$, which gives Eq. 67.

Finally, for $\hat{x}_0 = f^\star = \mathbb{E}[X_0 \mid x_t, t, y]$, the conditional Jensen/Cauchy–Schwarz inequality implies

$$\mathbb{E}[\|X_0 - f^\star\|_2 \mid x_t, t, y] \leq \sqrt{\mathbb{E}[\|X_0 - f^\star\|_2^2 \mid x_t, t, y]} = \sqrt{\mathrm{tr}(\mathrm{Cov}(X_0 \mid x_t, t, y))}, \tag{78}$$

which is Eq. 68. This makes explicit that posterior concentration implies a small plug-in error. □

### A.3.5. THE CAUCHY CONSISTENCY LOSS

**Negative log-likelihood.** Given a training triple $(x_0, y)$ and an intermediate sample $x_t \sim q(\cdot \mid x_0, y, t)$, we encourage *self-consistency* by maximizing the likelihood of $x_t$ under the induced bridge (or equivalently minimizing the NLL):

$$-\log q_\theta^{\mathrm{ind}}(x_t \mid y, t) = \sum_{i=1}^{d} \Big( \log \pi + \log \big((x_{t,i} - \hat{\mu}_{t,i})^2 + \gamma_{t,i}^2\big) - \log \gamma_{t,i} \Big). \tag{79}$$

When $\gamma_{t,i}$ is fixed (schedule-controlled) and independent of $\theta$, the terms $\log \pi - \log \gamma_{t,i}$ are constants w.r.t. $\theta$ and can be dropped. This yields the Cauchy consistency loss used in the main text:

$$\mathcal{L}_{\mathrm{cauchy}} \propto -\log q_\theta^{\mathrm{ind}}(x_t \mid y, t) = \sum_{i=1}^{d} \log\Big((x_{t,i} - \hat{\mu}_{t,i})^2 + \gamma_{t,i}^2\Big). \tag{80}$$

**Robustness: bounded influence of the Cauchy score.** The Cauchy bridge is particularly suitable for heavy-tailed residuals due to the bounded influence of its score. For each coordinate, from Eq. 59,

$$\left| \frac{\partial}{\partial x_{t,i}} \log q(x_t \mid x_0, y) \right| = \frac{2|z_i|}{z_i^2 + \gamma_t^2} \leq \frac{1}{\gamma_t}, \qquad z_i \triangleq x_{t,i} - \mu_{t,i}, \tag{81}$$

with the maximum attained at $|z_i| = \gamma_t$. Consequently,

$$\left\| \nabla_{x_t} \log q(x_t \mid x_0, y) \right\|_2 \leq \frac{\sqrt{d}}{\gamma_t}. \tag{82}$$

Where $d$ denotes the signal dimensionality. Unlike the Gaussian score whose magnitude grows linearly with $\|x_t - \mu_t\|$, the Cauchy score is *bounded* and *redescending*: it reaches its maximum at $|z_i| = \gamma_t$ and decays as $O(1/|z_i|)$ for large residuals. Therefore, extreme impulsive corruptions (e.g., speech bursts) cannot induce arbitrarily large drifts/gradients, which stabilizes both training and sampling dynamics under heavy-tailed bowel sounds denoising.

### A.4. Validity of Cauchy-DBIM sampling via Gaussian scale mixtures

**Lemma 1.** [Cauchy as a Gaussian scale mixture] Let $z, u \sim \mathcal{N}(0, 1)$ be independent. Then $\xi = z/u$ follows a standard Cauchy distribution. Equivalently, since $\mathrm{sign}(u)$ is independent of $|u|$ and can be absorbed into $z$, we may write $\xi \stackrel{d}{=} z/|u|$. For numerical stability, we use the regularized form $\xi_\varepsilon = z/(|u| + \varepsilon_u)$ with $\varepsilon_u > 0$, which converges in distribution to Cauchy$(0, 1)$ as $\varepsilon_u \to 0$. Thus a (regularized) Cauchy noise with per-coordinate scale $s_i$ can be generated as

$$s_i \xi_{\varepsilon,i} = \kappa_i z_i, \qquad \kappa_i \triangleq \frac{s_i}{|u_i| + \varepsilon_u}, \quad z_i \sim \mathcal{N}(0, 1). \tag{83}$$

Conditioned on $u$, this is Gaussian with random variance: $s_i \xi_{\varepsilon,i} \mid u_i \sim \mathcal{N}(0, \kappa_i^2)$.

**Proposition 2.** [Conditional Gaussian bridge and DBIM reuse] Consider the Cauchy-driven bridge parameterization $x_t = \mu_t(x_0, y) + c_t \odot (s \odot \xi)$ with i.i.d. $\xi_i \sim \text{Cauchy}(0, 1)$. Using the scale-mixture reparameterization above, conditional on $u$ the residual becomes Gaussian:

$$x_t = \mu_t(x_0, y) + \hat{c}_t(u) \odot z, \qquad z \sim \mathcal{N}(0, I), \qquad \hat{c}_{t,i}(u) = c_t \, \kappa_i. \tag{84}$$

Hence, conditioned on $u$, the bridge reduces to a linear-Gaussian bridge with known effective scale $\hat{c}_t$. Therefore, any Gaussian-DBIM update that follows from linear-Gaussian conditioning remains valid coordinate-wise after replacing $c_t$ by $\hat{c}_t(u)$ and recomputing all derived quantities accordingly. In particular, when the Gaussian update uses a noise-injection scale $\rho_n$ that is degree-1 homogeneous in $\{c_{t_n}\}$, scales as $\hat{\rho}_{n,i} = \kappa_i \rho_n$.

*Proof sketch.* Using Lemma 1., write the Cauchy noise as $s \odot \xi = \kappa(u) \odot z$ with $z \sim \mathcal{N}(0, I)$ and $\kappa_i(u) = s_i/(|u_i| + \varepsilon_u)$. Conditioned on $u$, we have

$$x_t = \mu_t(x_0, y) + c_t \odot (\kappa(u) \odot z) = \mu_t(x_0, y) + \hat{c}_t(u) \odot z, \tag{85}$$

where $\hat{c}_{t,i}(u) = c_t \, \kappa_i(u)$. Hence $x_t \mid (x_0, y, u)$ is Gaussian with mean $\mu_t(x_0, y)$ and diagonal covariance $\text{diag}(\hat{c}_t(u)^{\odot 2})$. All Gaussian-DBIM updates are derived solely from linear-Gaussian conditioning with known (diagonal) scales, therefore they can be applied conditional on $u$ by replacing the scale schedule $c_t$ with $\hat{c}_t(u)$ (and recomputing any derived noise-injection scale via the same functional form). $\qquad \square$

**Corollary (Marginal heavy-tail correctness).** Let $\xi_{\varepsilon,i} = z_i/(|u_i| + \varepsilon_u)$ with $z_i, u_i \overset{\text{i.i.d.}}{\sim} \mathcal{N}(0, 1)$. If $(u, z)$ are resampled independently across timesteps (and coordinates), then for each step $n$ the injected residual $c_{t_n}(s_i \xi_{\varepsilon,i})$ has the prescribed *per-step* (regularized) Cauchy-like marginal law with scale $c_{t_n} s_i$. Moreover, as $\varepsilon_u \to 0$, this marginal converges to the exact $\text{Cauchy}(0, c_{t_n} s_i)$.

*Proof sketch.* By Lemma 1, for each coordinate $i$, $s_i \xi_{\varepsilon,i} \overset{d}{=} \kappa_i z_i$ with $\kappa_i = s_i/(|u_i| + \varepsilon_u)$ and $z_i \sim \mathcal{N}(0, 1)$. Therefore, $c_{t_n}(s_i \xi_{\varepsilon,i}) \overset{d}{=} c_{t_n} \kappa_i z_i$, and marginalizing out $(u_i, z_i)$ yields a (regularized) Cauchy-like distribution whose scale is $c_{t_n} s_i$ (becoming exact Cauchy as $\varepsilon_u \to 0$). Independently resampling $(u, z)$ across steps ensures the same marginal form at every timestep. $\qquad \square$

**Shared $u$.** In practice, we may sample $u$ once and reuse it along the entire reverse trajectory (shared $u$), while resampling only $z$ at each timestep. Conditioned on this shared $u$, the process becomes a *single* diagonal linear-Gaussian bridge with fixed effective scale $\hat{c}_t(u) = c_t \odot \kappa(u)$, enabling stable closed-form DBIM updates. Unconditionally, marginalizing out $u$ still induces heavy-tailed residual magnitudes, but now the effective noise scale is coupled across timesteps (introducing temporal dependence). Empirically, this variance-reduction often improves numerical stability compared to resampling $u$ at every step, while maintaining the intended heavy-tailed behavior in the residuals.

# B. Appendix: Dataset

## B.1. Detailed Breakdown of the CLINBS dataset

### B.1.1. DATA ACQUISITION AND DATASET CONSTRUCTION

All bowel sound recordings in the CLINBS dataset were collected using a custom-modified electronic stethoscope equipped with an internal high-sensitivity microphone. The acoustic signals were recorded in monaural format at a sampling rate of 48 kHz with a bit depth of 16 bits. To ensure stable signal quality, recordings were conducted in relatively quiet ward environments, with patients in a resting supine position. Each recording session lasted approximately 90 seconds, from which fixed-length segments were later extracted for dataset construction. To enhance data diversity and robustness to acquisition variability, bowel sounds were collected from multiple abdominal auscultation locations, including but not limited to the periumbilical area and different abdominal quadrants. The specific auscultation position was randomly selected across sessions rather than being fixed to a single anatomical location. During each recording, the stethoscope was placed directly on the skin surface and kept stable to minimize motion-induced artifacts.

A multi-stage quality control procedure was applied during both data acquisition and post-processing. During recording, sessions with excessive environmental noise, frequent patient movement, or unstable stethoscope placement were excluded.

*Table 7.* Patient Metadata and Data Usage Consent Form (Template)

| Item | Response Options |
| --- | --- |
| **Gender** | ☐ Female    ☐ Male    ☐ Prefer not to say |
| **Age Group** | ☐ ≤18    ☐ 19–40    ☐ 41–65    ☐ >65    ☐ Prefer not to say |
| **Primary Gastrointestinal Condition** | ☐ Abdominal infection / intestinal ischemia
☐ Intestinal obstruction / motility disorder
☐ Reduced gut function / intolerance
☐ Other |
| **Consent to Record Bowel Sounds** | ☐ Yes    ☐ No |
| **Consent to Use Data for Research Purposes** | ☐ Yes    ☐ No |
| **Consent to Release De-identified Metadata** | ☐ Yes    ☐ No |
| **Consent to Release De-identified Audio Data** | ☐ Yes    ☐ No |
| **Additional Notes (Optional)** | |

After acquisition, audio segments were further screened to remove samples with severe motion artifacts, prolonged silence, or dominant non-bowel sound interference. Ambiguous or low-quality recordings were reviewed and excluded based on consensus among clinicians and researchers to ensure the overall reliability of the dataset. Abnormal bowel sounds were defined based on clinically recognized acoustic patterns, including gas–liquid gurgles and metallic transients. These patterns were identified according to standard gastrointestinal auscultation criteria and independently annotated by multiple experienced gastroenterologists. To ensure label reliability, all abnormal sound annotations underwent a cross-validation process among experts, and labels were finalized only when consensus was reached.

To ensure a fair and clinically realistic evaluation, the CLINBS dataset was split into training, validation, and test sets following a patient-independent protocol. Specifically, recordings from the same patient were assigned exclusively to a single subset to avoid data leakage. The dataset was divided into 70% for training, 20% for validation, and 10% for testing. This splitting strategy enables a reliable assessment of model generalization to unseen patients.

### B.1.2. PATIENT METADATA AND DATA ANONYMIZATION

With appropriate authorization, we extracted patient metadata (Tab. 7) from available clinical records, including information on gender, age group, and pathological categories. In accordance with privacy protection requirements and patient preferences, a subset of samples does not include gender or age annotations. Throughout the dataset construction process, strict data anonymization procedures were applied. All personally identifiable information (PII), including patient names, identification numbers, exact dates, and location-specific metadata, was removed prior to analysis and storage. The released dataset contains only de-identified bowel sound recordings and high-level categorical annotations, ensuring that no individual patient can be re-identified. Importantly, all speech signals used in noise-interference experiments were sourced exclusively from the publicly available speech dataset. No patient speech recordings were collected or used at any stage of the dataset construction or experimental evaluation.

The study protocol was reviewed and approved by the institutional review board (IRB), and all procedures comply with applicable ethical guidelines for clinical data research. Dataset usage is restricted to research purposes, and access is subject to ethical and legal review.

### B.1.3. DETAILED DISTRIBUTION OF THE CLINBS DATASET

**Gender Distribution** The CLINBS dataset includes a diverse gender representation, though a significant portion of the data does not have a specified gender. Out of the 1531 samples, 36.07% are from female patients, and 33.93% are from male patients. A notable 30% of the samples do not have gender annotations.

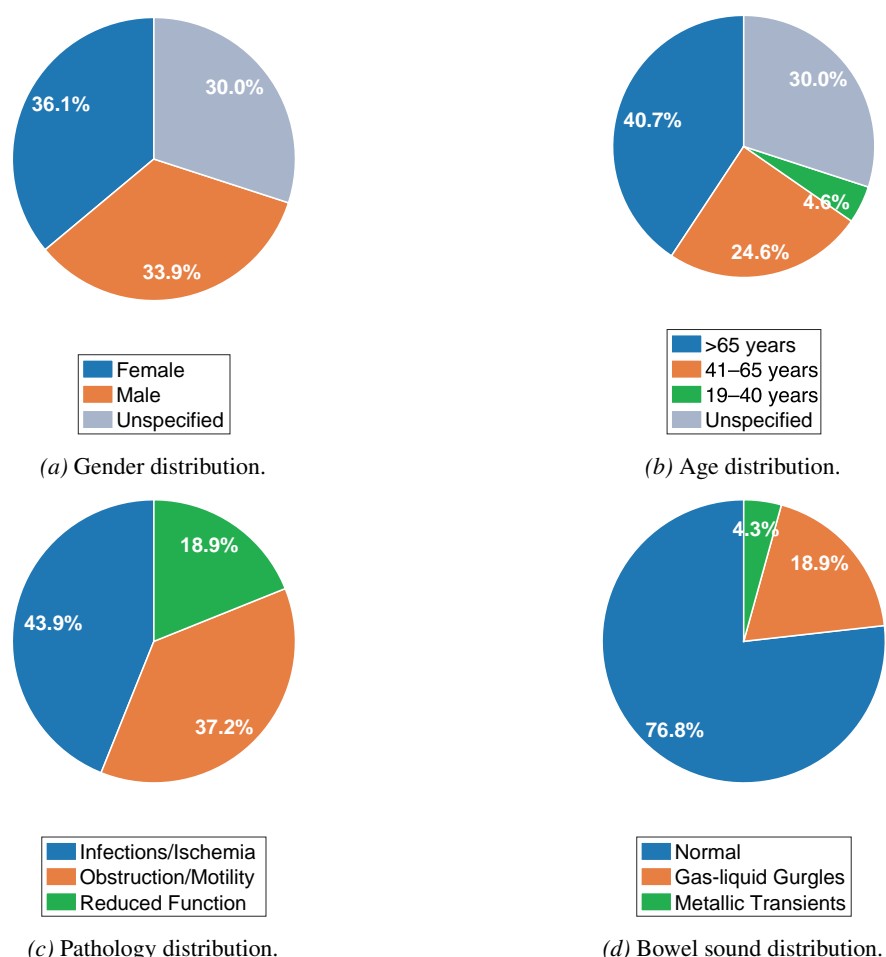

*(a)* Gender distribution.

*(b)* Age distribution.

*(c)* Pathology distribution.

*(d)* Bowel sound distribution.

*Figure 4.* Detailed distribution of the CLINBS dataset.

**Age Distribution**    The age distribution within the CLINBS dataset reveals that the majority of the samples come from patients over 65 years of age, accounting for 40.71% of the total dataset. The second-largest group consists of patients aged between 41 and 65 years (24.64%). A smaller portion, 4.64%, includes patients between 19 and 40 years of age. A large portion, 30%, of the dataset does not have age annotations. This age distribution can provide insights into the variations in bowel sounds across different age groups, with older patients likely experiencing different types of pathologies than younger individuals.

**Gastrointestinal Disease Categories**    The CLINBS dataset spans a wide range of gastrointestinal diseases, providing valuable data for various types of pathology detection. The majority of the samples (43.92%) represent cases of abdominal infections or intestinal ischemia, which are conditions commonly associated with abnormal bowel sounds. The second-largest category (37.16%) includes patients with intestinal obstruction or motility disorders, such as intussusception or abdominal hernia, both of which can generate distinct acoustic patterns. The remaining 18.92% of the samples are from patients with reduced gut function or intolerance, which could lead to different acoustic manifestations in bowel sounds.

**Abnormal Sound Breakdown**    The CLINBS dataset includes two key abnormal bowel sound patterns: gas-liquid gurgles (18.93%) and metallic transients (4.29%). Gas-liquid gurgles, typically associated with conditions like intestinal obstruction and gastrointestinal infections, are the more common abnormal sound and indicate significant pathologies such as intestinal ischemia. In contrast, metallic transients, though less frequent, are crucial for diagnosing severe conditions like bowel perforation or high-level obstruction. The inclusion of these rare but clinically significant sounds enhances the dataset's value, providing an essential resource for developing models that can detect both common and rare pathological events.

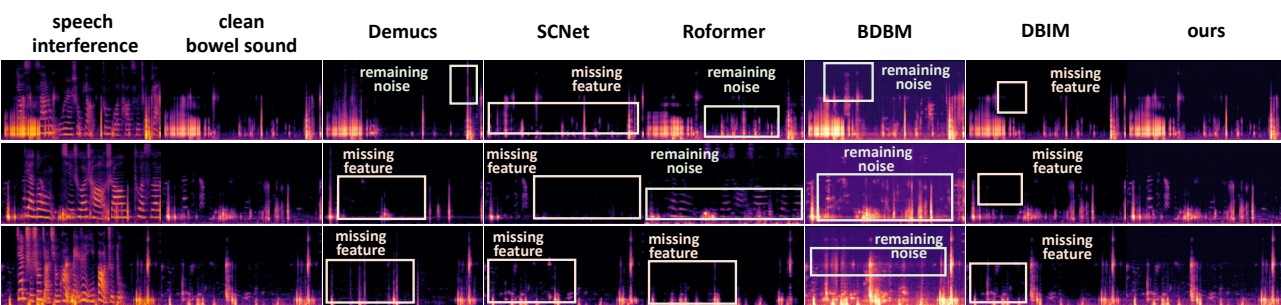

*Figure 5.* Comparison of different SOTA methods under speech interference.

### B.2. Speech Dataset: VCTK and AISHELL

**AISHELL (Fu et al., 2021)** AISHELL is a large-scale Mandarin speech corpus totaling roughly 1,000 hours. Its utterances span 12 application-oriented scenarios (e.g., wake words, command-and-control, smart home, autonomous driving, and industrial settings), offering diverse linguistic content for constructing speech interference. The data were collected in quiet indoor conditions with three recording devices operated in parallel, including a studio-grade microphone (44.1 kHz/16-bit) and Android/iOS smartphones (16 kHz/16-bit); AISHELL-2 releases the iOS-recorded channel for experiments. In total, 1,991 speakers from a broad range of accent regions in China participated, and the official transcription quality is reported to exceed 96% after professional annotation and strict verification. The corpus is distributed for academic research use, while commercial usage requires explicit permission from the dataset provider.

**VCTK (Yamagishi et al., 2019)** The CSTR VCTK Corpus contains read English speech from 110 speakers representing a wide variety of regional accents. Each speaker contributes on the order of 400 utterances, including sentences drawn from newspaper articles as well as two shared passages (the Rainbow Passage and an elicitation paragraph commonly used in accent resources); the per-speaker newspaper prompts are selected to improve contextual and phonetic coverage. All recordings were captured using a standardized setup in a hemi-anechoic chamber at the University of Edinburgh, employing two microphones (an omni-directional microphone and a wide-band condenser microphone) at 96 kHz/24-bit, followed by conversion to 16-bit and downsampling to 48 kHz with manual endpointing. A small number of recordings exhibit known technical exceptions (e.g., the wide-band microphone track for certain speakers), and the transcript for speaker `p315` is unavailable due to data loss. While originally released to support speaker-adaptive HMM-based TTS research, VCTK has since become a common benchmark for multi-speaker neural TTS and neural waveform modeling.

## C. Appendix: Evaluation

### C.1. Speech Interference on Bowel Sound

To visually assess the model's efficacy in restoring the target signal, Fig. 5 presents a comparative visualization of spectrograms across various methods. In the mixed signal, the target bowel sound is severely masked by human speech interference, exhibiting substantial spectro-temporal overlap in the frequency domain.

From the qualitative comparison, baseline models such as Demucs and Mel-Roformer can suppress part of the interference, yet they remain challenged by vocal harmonics that overlap with the bowel-sound band. Demucs typically leaves noticeable residual interference, while SCNet and Mel-Roformer show varying degrees of over-smoothing, which weakens fine-grained transients and fragments high-frequency harmonic structures. In contrast, our Cauchy-driven bridge model produces cleaner spectrograms with better-preserved textures and transient patterns, yielding reconstructions that most closely resemble the clean reference. These observations are consistent with the overall quantitative trends in Tab. 1, indicating that our method better balances interference suppression and signal fidelity under complex mixtures (examples are shown in Fig. 6).

### C.2. Robustness against Speech Interference

To evaluate the framework's universal efficacy against non-stationary interference, we conducted the evaluation using two linguistically distinct datasets: VCTK (English) and AISHELL (Mandarin). Interference was simulated by linearly mixing clean bowel sounds with speech at ratios of $\lambda \in \{0.3, 0.5, 0.7\}$ (Tab. 8). Each model was trained for 1000 iterations under

*Table 8.* Robustness against Speech Interference.

| Metric | VCTK ($\lambda$) | | | AIShell ($\lambda$) | | |
|---|---|---|---|---|---|---|
| | 0.3 | 0.5 | 0.7 | 0.3 | 0.5 | 0.7 |
| MAE $\downarrow$ | **5.9** | 8.8 | 11.8 | **6.3** | 9.0 | 12.7 |
| PSNR $\uparrow$ | **26.39** | 23.23 | 22.89 | **26.12** | 23.11 | 22.55 |
| PCC $\uparrow$ | **0.949** | 0.928 | 0.907 | **0.944** | 0.919 | 0.884 |
| SSIM $\uparrow$ | **0.791** | 0.652 | 0.589 | **0.752** | 0.618 | 0.582 |
| LPIPS $\downarrow$ | **0.122** | 0.206 | 0.262 | **0.159** | 0.217 | 0.271 |
| LSD $\downarrow$ | **9.69** | 13.74 | 17.66 | **10.97** | 16.99 | 18.70 |

*Table 9.* Effect of training diffusion steps $T_{\text{train}}$.

| $T_{\text{train}}$ | FID $\downarrow$ | IS $\uparrow$ |
|---|---|---|
| 200 | 55.27 | 2.06 |
| 500 | 41.05 | 2.08 |
| 700 | 39.32 | 2.12 |
| **1000** | **38.54** | **2.13** |

*Table 10.* Effect of sampling steps $T_{\text{sample}}$.

| $T_{\text{sample}}$ | FID $\downarrow$ | IS $\uparrow$ |
|---|---|---|
| 10 | 63.15 | 2.08 |
| **20** | **38.54** | **2.13** |
| 50 | 44.10 | 2.09 |
| 100 | 47.69 | 2.08 |

identical settings. The model maintains high structural fidelity (SSIM) and correlation (PCC) even under severe masking ($\lambda = 0.7$). This cross-linguistic resilience demonstrates that the framework learns fundamental signal-noise separation rather than language-specific patterns. While VCTK reconstruction is slightly more robust due to the spectral sparsity of English phonemes compared to Mandarin's continuous tonal flow.

### C.3. Sensitivity Analysis of Diffusion-Step Parameters

To study the effect of diffusion-step parameters on generation quality, we conduct a parameter sensitivity analysis on both the training diffusion horizon $T_{\text{train}}$ and the inference budget $T_{\text{sample}}$. All other hyperparameters are kept fixed , and each configuration is trained for 1000 iterations. Specifically, we train models with $T_{\text{train}} \in \{200, 500, 700, 1000\}$. For sampling, we use a model trained with a fixed $T_{\text{train}}$ and vary the number of reverse steps as $T_{\text{sample}} \in \{10, 20, 50, 100\}$. The results demonstrate that generation quality improves consistently with the increase of training steps $T_{\text{train}}$, reaching peak performance at 1000 steps with the lowest FID (38.54) and highest IS (2.13). Regarding inference efficiency, the model achieves its optimal quality with only 20 sampling steps $T_{\text{sample}}$.

### C.4. Evaluation Under Naturally Recorded Clinical Noise

The additive observation model $y = x + n$ used in our main experiments is a first-order approximation. In real clinical recordings, the captured signal may also reflect room acoustics, stethoscope resonance, and body-conducted vibrations, which deviate from a purely additive assumption. To assess robustness under more realistic conditions, we conducted an additional experiment using naturally recorded ward noise. Under the same acquisition setup as CLINBS, we selected segments verified by clinicians to contain no bowel sound activity, thereby capturing ambient ward interference including reflected room acoustics, stethoscope coupling effects, and body-conducted vibrations. These segments were mixed with clean bowel sound samples at $\lambda = 0.5$, and evaluated using the same model configuration trained for 1,000 iterations.

As shown in Tab. 11, performance degrades only moderately compared to the synthetic speech corpora (VCTK and AISHELL), indicating that the framework remains reasonably robust even when interference is drawn from a realistic ward environment where the additive approximation is less idealized.

*Table 11.* Denoising performance under synthetic speech vs. naturally recorded clinical noise ($\lambda = 0.5$).

| Noise Source | PSNR $\uparrow$ | SSIM $\uparrow$ | LPIPS $\downarrow$ | PCC $\uparrow$ | MAE $\downarrow$ | LSD $\downarrow$ |
|---|---|---|---|---|---|---|
| VCTK | 23.23 | 0.652 | 0.206 | 0.928 | 8.8 | 13.74 |
| AISHELL | 23.11 | 0.618 | 0.217 | 0.919 | 9.0 | 16.99 |
| Clinical | 22.63 | 0.612 | 0.225 | 0.872 | 11.1 | 18.21 |

## C.5. Computational Cost Analysis

We report the model's computational profile in Tab. 12. On an NVIDIA RTX 4090, our framework processes a 5-second audio clip in 0.54 s with 20 reverse steps, which is approximately $9.3\times$ faster than real-time and sufficient for the intended clinical deployment on workstation-grade devices. We acknowledge that CPU and edge-device deployment would require further optimization such as quantization or pruning, which we consider an important direction for future work.

*Table 12.* Computational cost of the proposed framework.

| Metric | Value |
| --- | --- |
| Parameters | 553.8 M |
| FLOPs (per input) | 22,076.5 G |
| Inference latency | 0.54 s / 5 s clip |
| Hardware | NVIDIA RTX 4090 |
| Sampling steps | 20 |

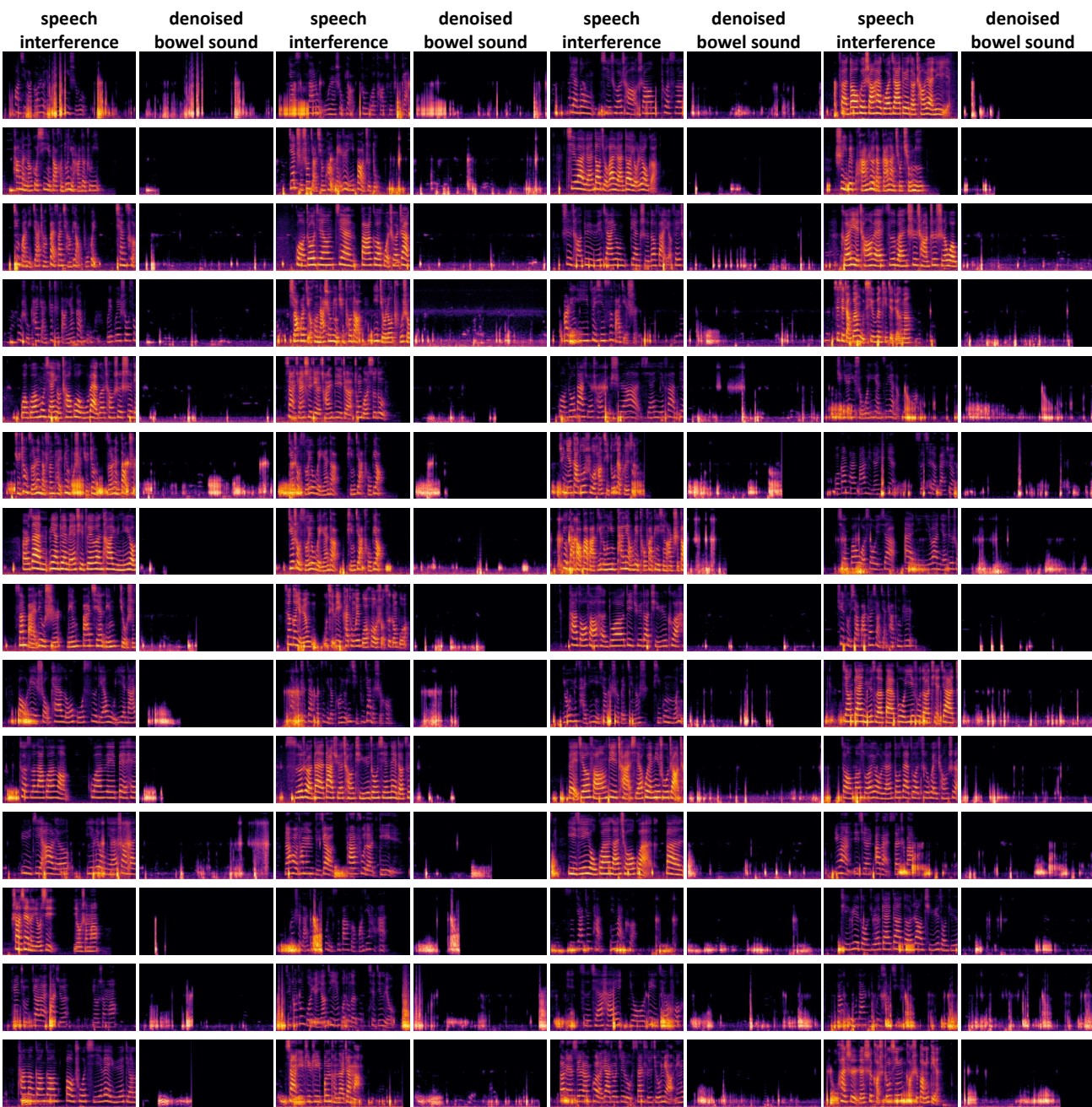

*Figure 6.* Mel-spectrogram visualization of bowel sound denoising with our method.

