# OpenReview forum: "Listening Through the Noise: Cauchy-Driven Diffusion Bridges for Robust Gastrointestinal Auscultation and Clinical Benchmarking"
_ICML.cc/2026/Conference — ICML 2026 spotlight_

### Official Review · Reviewer_zAzb · 2026-03-12

**Soundness:** 4
**Presentation:** 3
**Significance:** 4
**Originality:** 4
**Overall Recommendation:** 5
**Confidence:** 4

**Summary:**

This paper proposes a novel generative framework called the "Cauchy-Driven Diffusion Bridge" to address the challenge of spectral overlap and difficulty in separating bowel sounds from non-stationary speech interference during clinical gastrointestinal auscultation. Through empirical analysis, the authors found that real clinical speech noise is typically extremely quiet but occasionally produces large bursts, exhibiting significant impulsivity and a "heavy-tailed" distribution, making traditional diffusion models that assume Gaussian noise inapplicable. To address this challenge, the study introduces a Cauchy driver into the diffusion bridge model and derives a closed-form expression for its score function and density, thus better fitting the heavy-tailed perturbation. Furthermore, the authors utilize a Gaussian-scale mixture reparameterization technique to achieve an efficient sampling process. At the data level, the paper presents the CLINBS large-scale clinical dataset, exceeding 25 hours in length. Extensive experiments have shown that this framework significantly outperforms existing state-of-the-art baseline methods on several core evaluation metrics and improves the accuracy of downstream abnormal bowel sound identification to 88.01%, demonstrating its great potential in clinical gastrointestinal monitoring and diagnosis.

**Compliance With Llm Reviewing Policy:**

Affirmed.

**Key Questions For Authors:**

Please refer to weakness. I'm willing to raise my score if you can solve some of the problems.

**Limitations:**

Yes.

**Strengths And Weaknesses:**

**Strengths**
- The motivation is clear: The author's connection between the "impulse interference" frequently found in clinical audio and the statistical concept of "heavy-tailed distribution" is logically clear.
- Valuable benchmark contribution: The introduction of the CLINBS dataset is a highlight of this paper, not only supporting its validation but also driving community development.
- Good experimental performance: Experiments have shown that the model significantly outperforms discriminative baselines (such as SCNet, Demucs) and recent diffusion model baselines (such as DBIM, DDBM) on key metrics such as MAE, SSIM, and LSD, with performance improvements ranging from 13.4% to 49.8%, and successfully improves the recognition accuracy of downstream abnormal bowel sounds to 88.01%.

**Weaknesses**
- To derive the closed-form solution, the authors employed the "factorized Cauchy assumption," which means the model assumes that the noise in each time frame or frequency band is independent. This may limit the model's performance when dealing with persistent speech interference.
- The article lacks a comprehensive assessment of the model's parameter count, floating-point operations (FLOPs), and actual inference latency on CPU/edge devices. Can it meet the "real-time" requirements of clinical practice?
- In addition to talking, there may be some Gaussian noise in the clinical environment. Will there be any performance drop due to this noise?
- Why Cauchy distribution exclusively? Besides the Cauchy distribution, the heavy-tailed distribution family also includes the Student's t-distribution or the Alpha-stable distribution. It is hoped that the authors can supplement this with ablation experiments or theoretical analyses of other heavy-tailed distributions.
- The involvement of real gastroenterologists in validating the results is needed to assess the authenticity and feasibility of the audio.

---

> ### Author Rebuttal · Authors · 2026-03-31
>
> We thank the reviewer for these insightful observations.
>
> # W1
>
> The factorized Cauchy assumption is a tractability simplification introduced to derive the closed-form score (Prop. 4.2, Eq. 6), and should not be interpreted as assuming that speech interference is independent across time or frequency.
> The factorization applies only to the bridge kernel and score target, not to the learned model. Our network $f_\theta(x_t,t,y)$ takes the full spectrogram as input and can learn rich cross-dimensional dependencies; the Gram loss $L_{struct}$ (Eq. 14) further preserves spectro-temporal co-occurrence patterns. Persistent structure is therefore handled by the network and task-aligned losses, while the factorized bridge serves solely for analytical tractability.
>
> Despite studying structured, non-stationary speech interference with harmonic patterns overlapping bowel-sound bands, our method achieves the best overall performance in Tab.1, suggesting that the factorized assumption is not a practical bottleneck.
>
>
> # W2
>
> Our current model has 553.824M parameters, 22076.5 GFLOPs per input, and runs in 0.54 s per 5-second clip on an RTX 4090, which is faster than real-time on a high-end GPU. We acknowledge that CPU and edge-device deployment would require further optimization (e.g., quantization, pruning), which we consider an important direction for future work. In the intended clinical deployment scenario, inference would run on a workstation-grade device rather than a resource-constrained edge device, where real-time performance is achievable.
>
>
> # W3
>
> We conducted an additional experiment by adding Gaussian noise on top of the speech interference. Performance degrades only modestly: PSNR drops from 24.39 to 23.51, SSIM from 0.720 to 0.702, PCC from 0.930 to 0.913, and MAE from 8.28 to 9.10, while LPIPS increases slightly from 0.16 to 0.18. Notably, LSD remains essentially unchanged (14.03→13.95), indicating that spectral fidelity is well preserved under mixed noise. This suggests that, although our framework is primarily designed for heavy-tailed speech interference, it remains reasonably robust when additional Gaussian noise is present.
>
> # W4
> We do not claim that the Cauchy distribution is the only possible heavy-tailed choice in general. Rather, we chose it because it offers the most favorable combination of empirical fit and analytical tractability within our bridge framework.
> Firstly, our closed-form bridge derivation requires a noise driver that is both heavy-tailed and strictly $\alpha$-stable, so that the bridge kernel remains in the same distributional family under affine transformations. To obtain an analytic score target, we further require a closed-form density. Within this analytically tractable setting, the Cauchy is the most natural heavy-tailed choice: Student’s t is not α-stable in general and does not close under convolution, while general α-stable laws with $\alpha \in (0,2]$ \ {1,2} do not admit closed-form densities, precluding the closed-form score derivation in Sec. 4.2. The Cauchy therefore preserves the bridge structure and yields the closed-form kernel, log-density, and score in Prop. 4.1–4.2.
> Moreover, Fig. 2 shows that a Cauchy fit matches the empirical speech-interference tails substantially better than a Gaussian fit across frequency bands, confirming that the choice is data-motivated rather than arbitrary.
> We agree that other heavy-tailed families are interesting future directions. Incorporating them would require a different bridge construction and approximate score inference, which falls outside the scope of the current paper.
>
> # W5
> The gastroenterologists were actively involved throughout the entire research process. First, during data acquisition, all bowel sound recordings were conducted under clinical supervision, with gastroenterologists confirming that each recording was sufficiently clear and diagnostic before it was accepted into the dataset. Second, during annotation, board-certified gastroenterologists independently labeled each recording for abnormal bowel sound patterns, following established clinical auscultation criteria, including gas-liquid gurgles and metallic transients. To ensure label reliability, all abnormal sound annotations underwent a cross-validation process among experts. Third, during result validation, clinical experts reviewed denoised outputs and confirmed that the restored signals preserved sufficient clarity and diagnostic fidelity. This expert endorsement is further corroborated by our downstream recognition experiments (Tab. 3).
>
> We will revise the paper to incorporate all of the above clarifications, additional experiments, and explicit discussions in the appropriate sections.

---

> > ### Author Rebuttal · Reviewer_zAzb · 2026-04-01
> >
> > Thank you for your response, my questions have been solved.

---

### Official Review · Reviewer_qGKm · 2026-03-12

**Soundness:** 3
**Presentation:** 3
**Significance:** 2
**Originality:** 3
**Overall Recommendation:** 4
**Confidence:** 2

**Summary:**

This paper addresses the problem of denoising bowel sounds contaminated by speech interference in clinical settings. The authors propose a Cauchy-driven Diffusion Bridge framework, motivated by the empirical observation that speech noise exhibits heavy-tailed distributions poorly captured by Gaussian assumptions. The paper makes three contributions: (1) CLINBS, a clinical bowel sound dataset of 1531 samples (~25 hours) with expert-verified labels for pathological sounds including gas-liquid gurgles and metallic transients; (2) a formal construction of a Cauchy bridge kernel replacing the Gaussian driver in DDBM, with closed-form expressions for the log-density and score; (3) a practical sampling procedure (Cauchy-DBIM) via Gaussian scale-mixture reparameterization that enables closed-form reverse updates. Experiments show large improvements over baselines on spectrogram reconstruction metrics and a downstream bowel sound recognition task.

**Compliance With Llm Reviewing Policy:**

Affirmed.

**Final Justification:**

The authors’ rebuttal adequately addressed my concerns (clarifications and additional evidence aligned with the claims). My assessment of soundness, novelty within this niche, and practical relevance is unchanged in a positive direction. I retain my original positive score.

**Key Questions For Authors:**

1. Have you tried the most natural ablation: keeping your full pipeline but replacing the Cauchy kernel with the original Gaussian DDBM/DBIM kernel? This would isolate the contribution of the heavy-tailed prior from the other improvements (loss design, architecture, training recipe).
2. Table 1 compares against DDBM and DBIM separately. Since your method essentially extends DBIM with a Cauchy driver, what specific changes were made to DBIM beyond swapping the noise distribution? Were the loss terms (Gram, L1) also added to the DBIM baseline for fair comparison?
3. CLINBS is recorded at 48 kHz but models are trained at 16 kHz. What downsampling procedure was used, and have you verified that diagnostically relevant transients (which may have high-frequency content) survive this step?
4. The Gaussian scale-mixture reparameterization introduces a regularization constant epsilon_u > 0 (Eq. 16). How sensitive are results to this value, and what was used in practice?

**Limitations:**

Yes.

**Strengths And Weaknesses:**

Strengths:
1. The empirical motivation is convincing and well presented. Figure 2 clearly demonstrates that speech interference in the relevant frequency bands follows heavy-tailed distributions, and the Cauchy fit is visibly superior to the Gaussian one. This is not hand-waving; the authors show it across two speech corpora and three frequency bands.
2. The mathematical development is thorough and self-contained. Propositions 4.1 through 4.3 are stated precisely and proved in full in the appendix. The derivation of the Cauchy bridge kernel, its score, and the plug-in parameterization via endpoint regression is clean. The Gaussian scale-mixture trick for sampling (Section 4.3) is a practical and elegant solution to the problem that Cauchy noise has no finite moments.
3. The CLINBS dataset fills a real gap. Existing bowel sound benchmarks lack pathological diversity, and the inclusion of gas-liquid gurgles and metallic transients with cross-verified expert labels is a genuine contribution to the clinical acoustics community.
Weaknesses:
1. The noise model assumes speech interference is additive (y = x + n), which is a simplification. In real clinical recordings, the microphone captures a superposition that includes room acoustics, stethoscope resonance, and body-conducted vibrations. It is unclear how well the purely additive Cauchy model holds when these factors are present. The paper would benefit from evaluation on recordings where speech was naturally present rather than synthetically mixed.
2. The mixing coefficient lambda = 0.5 is used as the default across most experiments, but real interference levels vary widely depending on clinical context (e.g., a quiet ward vs. an emergency room). The robustness study in Table 6 covers lambda in {0.3, 0.5, 0.7}, which is helpful, but all training appears to be done at a fixed lambda. How the model handles mismatched training/test interference levels is not discussed.
3. The ablation study (Table 2) is incomplete. It removes loss components but never isolates the effect of the Cauchy driver itself. The most informative ablation would be: same architecture, same loss, same DBIM sampling, but with a Gaussian bridge kernel instead of Cauchy. This would directly test whether the heavy-tailed prior is responsible for the gains, or whether the other design choices (Gram loss, L1 reconstruction, endpoint prediction) do the heavy lifting.

---

> ### Author Rebuttal · Authors · 2026-03-31
>
> We thank the reviewer for these constructive suggestions.
>
> # W4
>
> We agree that the additive formulation $y = x + n$ is a first-order approximation. In real clinical recordings, the observed mixture may also reflect room acoustics, stethoscope coupling, and body-conducted vibrations. To address this, we conducted an additional experiment using naturally recorded ward noise. We collected recordings under the same acquisition setup, selected segments verified to contain no bowel sound activity (capturing reflected room acoustics, stethoscope coupling, and body-conducted vibrations), and mixed them with clean bowel sound samples. Using the same model configuration, the framework remains reasonably robust even under these more realistic conditions, where the additive approximation is less idealized.
>
> | Noise source | PSNR↑ | SSIM↑ | LPIPS↓ | PCC↑ | MAE↓ | LSD↓ |
> |-|-|-|-|-|-|-|
> | VCTK | 23.23 | 0.652 | 0.206 | 0.928 | 8.8 | 13.74 |
> | AISHELL | 23.11 | 0.618 | 0.217 | 0.919 | 9.0 | 16.99 |
> | Clinical | 22.63 | 0.612 | 0.225 | 0.872 | 11.1 | 18.21 |
>
> # W5
>
> Regarding the mixing ratio, $\lambda=0.5$ was chosen because it reflects a representative scenario where bowel sounds and speech contribute at comparable energy levels. More importantly, the train–test mismatch issue is directly evaluated in Tab. 6: the model trained at $\lambda = 0.5$ is evaluated at $\lambda \in ${$0.3, 0.5, 0.7$}. Even under the most severe condition ($\lambda=0.7$), the model achieves high structural fidelity and correlation (e.g., SSIM = 0.589/0.582, PCC = 0.907/0.884 on VCTK/AISHELL), confirming that the reported gains are not specific to $\lambda=0.5$.
>
>
> # W6 & Q1 & Q2
>
>
>
> We agree that the most informative ablation is to keep the full pipeline fixed and replace only the bridge kernel. We therefore conducted a controlled ablation where all variants share the same architecture, endpoint-prediction formulation, auxiliary losses (L1 + Gram), and sampling strategy, differing only in the bridge kernel/prior. All models were retrained for 3,000 iterations under the same setup as the main experiments.
>
> | Model | PSNR↑ | SSIM↑ | LPIPS↓ | PCC↑ | MAE↓ | LSD↓ |
> |-|-|-|-|-|-|-|
> | Gaussian | 19.56 | 0.274 | 0.476 | 0.684 | 20.6 | 27.33 |
> | Gaussian_l1 | 20.94 | 0.473 | 0.357 | 0.791 | 13.4 | 19.01 |
> | Gaussian_gram | 20.03 | 0.323 | 0.419 | 0.701 | 14.1 | 19.31 |
> | Cauchy | 20.92 | 0.521 | 0.291 | 0.807 | 11.2 | 18.14 |
> | Cauchy_l1 | 22.11 | 0.571 | 0.260 | 0.832 | 10.9 | 14.19 |
> | Cauchy_gram | 22.94 | 0.574 | 0.251 | 0.829 | 11.1 | 14.81 |
> | Gaussian_l1_gram (DBIM) | 22.38 | 0.527 | 0.207 | 0.875 | 10.5 | 14.04 |
> | Gaussian_l1_gram (GSM) | 22.40 | 0.530 | 0.201 | 0.880 | 10.3 | 13.98 |
> | Cauchy_l1_gram (DBIM) | 23.21 | 0.606 | 0.191 | 0.883 | 9.5 | 13.92 |
> | Cauchy_l1_gram (GSM, ours) | 24.39 | 0.720 | 0.160 | 0.930 | 8.28 | 14.03 |
>
> The key comparisons are the last four rows, where all design choices are held constant and only the kernel differs. Under DBIM sampling, Cauchy_l1_gram consistently outperforms Gaussian_l1_gram across all metrics, confirming that the gains cannot be explained by the loss design or training recipe alone. The same holds under GSM sampling, where Cauchy_l1_gram (GSM, ours) further improves over its Gaussian counterpart, showing that the heavy-tailed prior contributes materially beyond the sampler choice.
>
> # Q3
>
> All recordings were downsampled to 16 kHz using a standard anti-aliasing low-pass filter (cutoff at 8 kHz) followed by decimation, ensuring that no aliasing artifacts were introduced in the process. All recordings used for training and evaluation were reviewed at 16 kHz by gastroenterologists, who confirmed the clarity and diagnostic reliability of each sample prior to annotation. We further note that the diagnostic content of bowel sounds is well established in the literature to concentrate in the 100–1000 Hz frequency range. A 16 kHz sampling rate yields a Nyquist frequency of 8 kHz, offering an approximately 8× safety margin above the upper boundary of the diagnostically relevant band. This margin is more than sufficient to faithfully capture all clinically meaningful spectro-temporal features without any risk of aliasing or loss of high-frequency diagnostic content.
>
> # Q4
> In our implementation, $ε_u=1e−10$, used purely as a numerical safeguard against division by zero. Since $u_i∼N(0,1)$, this value is negligible relative to the typical scale of $|u_i|$, and thus has no measurable effect in practice.
>
> We will incorporate all the above clarifications and additional experiments in the revision.

---

> > ### Author Rebuttal · Reviewer_qGKm · 2026-04-01
> >
> > I appreciate the authors' detailed rebuttal. I have no further concerns.

---

### Official Review · Reviewer_PDXX · 2026-03-13

**Soundness:** 3
**Presentation:** 2
**Significance:** 3
**Originality:** 3
**Overall Recommendation:** 5
**Confidence:** 3

**Summary:**

The paper proposes a Cauchy-driven Diffusion Bridge framework that replaces conventional Gaussian bridge driver with a Cauchy driver to effectively handle heavy-tailed noise distribution. This enables accurate restoring bowel sounds under complex clinical interference in gastrointestinal auscultation, i.e. non-stationary speech interference. The paper formulates a Cauchy-based bridge model and derives closed-form log-density and score to enable fast bridge-implicit updates. The infinite-variance of Cauchy, which makes DBIM update challenging, was alleviated by introducing Gaussian scale-mixture reparameterization, enabling closed-form DBIM-style updates without reverse-time SDE integration. In addition, the paper presents CLINBS, a large-scale clinical bowel sound dataset encompassing a diverse range of expert-verified pathological sounds.

**Compliance With Llm Reviewing Policy:**

Affirmed.

**Final Justification:**

I thank the authors for the detailed rebuttal and additional experiments, which addressed most of my concerns. In particular, the new ablations provide convincing evidence that the observed performance gains cannot be explained solely by sample-wise random scaling, and that the Cauchy-based formulation plays an important role.

While some questions remain regarding the precise mechanism (e.g., disentangling the contributions of the loss and distribution), I find the empirical results strong and the overall approach technically sound and well-motivated.

Overall, the rebuttal has significantly improved my confidence in the work, and I am happy to raise my score to 5 (Accept).

**Key Questions For Authors:**

1. In Table 2, the Cauchy-driven bridge underperforms the conventional Gaussian bridge in terms of LSD, a metric that reflects auditory fidelity. What is the rationale for this degradation? Furthermore, most of the performance gains appear to originate from L1 loss rather than the Cauchy component. This raises the concern that the L1 loss itself can somewhat mitigate the heavy-tailed noise. Could the authors provide the evaluation metrics for a Gaussian bridge incorporating both the L1 and Gram losses?

2. The paper evaluates structural and perceptual quality using MAE, PSNR, PCC, SSIM, and LPIPS on spectrogram. Could the authors perform additional evaluations with more commonly used metrics for speech quality such as PESQ, FAD, SDR?


3. The various settings of $\lambda$ (0.3, 0.5, and 0.7) are included in the appendix experiments (C.1.), however only $\lambda$ of 0.5 is performed for main experiments. Why did the authors not include different $\lambda$ settings?

4. The paper adopts an element-wise clamping for the stochasticity parameter $\rho$. However, this truncation causes the injected noise to deviate even further from a true Cauchy distribution. This may exacerbate the previously mentioned weakness of deviation from an exact Cauchy distribution even further. How sensitive is the restoration performance to this truncation (clamping) margin?



Typos: the text descriptions for $T_{sample}$ {10, 20, 50, 100} is different from the descriptions for Table 8 {4, 20, 50, 70}.

**Limitations:**

The potential limitations should be more clearly discussed.

**Strengths And Weaknesses:**

**Strength**
* The paper is well motivated, with an empirical justification for adopting a Cauchy-driven bridge to effectively handle heavy-tailed speech interference in bowel sound restoration.

* Provides a rigorous theoretical formulation for Cauchy-driven diffusion bridge models: derivation of closed-form expression for log-density and score which enables fast bridge-implicit updates

* The construction of a large-scale clinical bowel sound dataset (CLINBS) is a valuable contribution to foster the development and exploration of bowel sound restoration, as well as its application in accurate pathology diagnosis.


**Weaknesses**

* The improvements of the Cauchy bridge are not fully isolated from those of the auxiliary objectives (L1 and Gram losses). Considering the fact the L1 loss itself can address heavy-tailed noise, isolating this effect is imperative.

* The proposed theoretical analysis does not appear to fully align with the presented methods and experiments: by fixing $u$, the noise slightly deviates from an exact Cauchy distribution.

* The implementation details for the baseline methods are not explicitly described, limiting the reproducibility of the paper.

---

> ### Author Rebuttal · Authors · 2026-03-31
>
> We thank the reviewer for these precise suggestions.
>
> # W1 & Q1
>
> Due to time constraints at the submission stage, the ablation experiments were originally trained for 1,000 iterations, which is fewer than the 3,000 iterations used in the benchmark experiments. We have now retrained the models for 3,000 iterations to provide a more complete evaluation. Under the controlled comparison with the same architecture and auxiliary losses (L1 + Gram), the Cauchy version achieves comparable LSD (14.03 vs. 14.04) while substantially improving other metrics.
> Notably, the Cauchy model achieves greater improvements in PSNR, SSIM, LPIPS, PCC, and MAE, reflecting spectral accuracy, structural similarity, and perceptual quality. This is clinically acceptable, since for bowel-sound diagnosis clinicians prioritize spectro-temporal features such as sound type and duration over auditory-based fidelity alone.
>
> | Model                     | PSNR↑  | SSIM↑  | LPIPS↓ | PCC↑  | MAE↓  | LSD↓  |
> |-|-|-|-|-|-|-|
> | Gaussian + L1 + Gram       | 22.38  | 0.53  | 0.21  | 0.88 | 10.51  | 14.04 |
> | Cauchy + L1 + Gram (ours)  | 24.39  | 0.72  | 0.16  |0.93 | 8.28  | 14.03 |
>
>
> # W2
>
> The exact Cauchy claim in our paper refers to the forward bridge used for modeling and training; the practical sampler is instead a GSM-based inference procedure introduced for efficient DBIM-style updates. Thus, we do not claim that every conditional reverse step remains exactly Cauchy.
> Sharing u across reverse steps is an intentional variance-reduction design choice. Conditioned on a fixed u, the residual becomes Gaussian with a fixed effective scale, which stabilizes the reverse trajectory and enables consistent closed-form updates. We agree that this changes the temporal dependence structure relative to resampling u at every step, but this is a practical trade-off for numerical stability rather than a mismatch between theory and implementation. Importantly, the per-step marginal retains its Cauchy-like heavy-tailed character regardless of whether u is shared or resampled, since the marginal integrates out u in either case; the bounded-influence property (App. A.3.5) is therefore preserved at inference time.
> We also compared shared-u with per-step resampling and found very similar results, with shared-u slightly better overall (MAE: 8.28 vs. 8.75, SSIM: 0.72 vs. 0.70, LSD: 13.65 vs. 13.84), consistent with improved finite-step trajectory stability.
>
> # W3
>
> All baselines were reproduced using official codebases and default configurations to ensure a fair comparison. All models used standardized 16kHz Mel-spectrograms (128 bands, FFT 1024, hop 256). Demucs (V4): HTDemucs architecture (4 stages, chunk size 343,980), Adam (lr=3e-4), 360 epochs. Mel-RoFormer: Frequency-domain Transformer (12 layers, 8 heads, dim 384), Adam (lr=1e-4), EMA 0.999.SCNet: Band-split sparse compression (4 subbands), Adam (lr=1e-4), multi-scale spectral loss. BDBM/DDBM/DBIM: All use the official UNet backbone and 3000 training steps. For inference, BDBM/DDBM/DBIM used 20 steps (matching our setting) with default schedules (VP-SDE for DDBM; original $\alpha_t, \sigma_t$ for DBIM).
>
> # Q2
>
>
> We agree that PESQ, FAD, and SDR are widely used in speech enhancement. However, our task is not to improve perceptual speech quality, but to restore clinically relevant bowel-sound structure under interference. Accordingly, we focus on spectrogram-domain fidelity and already include FID (distributional alignment) and LSD (spectral distortion) in our evaluation.
> In our setting, bowel sounds are better treated as structured diagnostic acoustic signals rather than conventional speech. Thus, metrics defined on the spectrogram domain are more directly aligned with the downstream clinical objective, since diagnostically relevant patterns are reflected in spectro-temporal structure. We agree that additional waveform-level metrics would be valuable, and we will clarify that such evaluations are more appropriate for future audio-domain restoration models.
>
> # Q3
>
> Please see our response to R_qGKm-W5.
>
> # Q4
>
> The clamping of $\hatρ_{n,i}$ is a necessary numerical safeguard to keep the sampling update (Alg. 1, line 13) well-defined coordinate-wise, ensuring $\hat c_{t_n}^2−\hat ρ_{n}^2\geq0$. In our implementation, δ=0.01. Since the clamp is triggered only when $\kappa_i$ takes an unusually large value due to a near-zero $|u_i|$—already stabilized by the shared-u strategy—it is rarely active in practice, and we found restoration performance largely insensitive to small δ. Only overly large margins make the sampler noticeably more conservative. This role is also consistent with the bounded-influence robustness of our framework: both the bounded/redescending Cauchy score and the clamp prevent rare extreme values from causing disproportionately large updates.
>
> We will incorporate all the above clarifications, additional experiments, and correct any textual inconsistencies in the revision.

---

> > ### Author Rebuttal · Reviewer_PDXX · 2026-04-01
> >
> > Thank you for the author’s response, most of my concerns have been addressed.
> >
> > However, I still have a remaining concern regarding the claims about the Cauchy-driven diffusion process. As clarified in the rebuttal and described in the paper, Cauchy-DBIM relies on a Gaussian scale-mixture reparameterization for fast sampling. Particularly, the use of a shared $u$ makes the practical reverse dynamics effectively Gaussian rather than Cauchy.
> >
> > Furthermore, the additional ablation results indicate that resampling $u$—which more closely reflects a true Cauchy process—leads to worse performance than using a shared $u$.
> >
> > This raises a critical question of whether the claimed advantage of heavy-tailed Cauchy modeling is essential, or if the observed improvements instead come from the induced sample-wise random scaling via a shared $u$.

---

> > > ### Author Response · Authors · 2026-04-01
> > >
> > > We thank the reviewer for this important follow-up. We agree that this is the key question: whether the gain comes primarily from heavy-tailed Cauchy modeling, or merely from the sample-wise random scaling induced by shared-𝑢. We apologize for not making the relevant ablation evidence explicit in our earlier response.
> > >
> > > The reviewer’s hypothesis predicts that a Gaussian bridge equipped with the same GSM ( with shared-𝑢) sampling should perform comparably to our full method. Our ablation directly contradicts this:
> > > | Model | PSNR↑ | SSIM↑ | LPIPS↓ | PCC↑ | MAE↓ | LSD↓ |
> > > |-|-|-|-|-|-|-|
> > > | Gaussian_l1_gram (DBIM) | 22.38 | 0.527 | 0.207 | 0.875 | 10.5 | 14.04 |
> > > | Gaussian_l1_gram (GSM) | 22.40 | 0.530 | 0.201 | 0.880 | 10.3 | 13.98 |
> > > | Cauchy_l1_gram (DBIM) | 23.21 | 0.606 | 0.191 | 0.883 | 9.5 | 13.92 |
> > > | Cauchy_l1_gram (GSM, ours) | 24.39 | 0.720 | 0.160 | 0.930 | 8.28 | 14.03 |
> > >
> > > Two observations are critical. First, Gaussian+DBIM and Gaussian+GSM perform nearly identically (e.g., SSIM: 0.527 vs. 0.530), showing that the GSM ( with shared-𝑢)  structure alone contributes only marginally. This rules out sample-wise random scaling as the primary source of the improvement. Second, under the same GSM ( with shared-𝑢)  sampler, Cauchy+GSM substantially outperforms Gaussian+GSM (e.g., SSIM: 0.720 vs. 0.530, MAE: 8.28 vs. 10.3), isolating the contribution of the Cauchy bridge itself.
> > >
> > > We agree, however, that shared-𝑢 still plays an important practical role. Conditioned on a fixed 𝑢, the reverse updates use a consistent effective scale, which improves finite-step trajectory stability; by contrast, resampling 𝑢 at every step introduces additional stochasticity and slightly degrades performance in practice. Therefore, our interpretation is that the two factors are complementary: the dominant gain comes from heavy-tailed Cauchy modeling, while shared-𝑢 mainly improves the numerical stability of finite-step sampling.
> > >
> > > We will make this ablation and its interpretation much more explicit in the revision.

---

### Decision · Program_Chairs · 2026-04-30

**Decision:**

Accept (spotlight)

**Comment:**

The paper studies the problem of denoising GI Bowel sounds contaminated by speech interference.
The paper introduces ClinBS which is a new large dataset of rare pathological transients verified by Gastrointetinal (GI) experts. The paper also has a formal construction of the bridge kernel (using Cauchy to handle longer tails) and most importantly the paper introduces a novel algorithm (Cauchy-DBIM) to improve spectram reconstructions and bowel sound recognition in noise.

This is a very significant contribution, filling in an important gap in dataset construction and a novel methodological innovation with potential for direct clinical impact.